# WHEN TWO IS ENOUGH: COT–POT ENSEMBLING FOR EFFICIENT SELF-CONSISTENCY IN LLM REASONING

## ABSTRACT

Self-consistency (SC) is a popular technique for improving the reasoning accuracy of large language models by aggregating multiple sampled outputs, but it comes at a high computational cost due to extensive sampling. We introduce a hybrid ensembling approach that leverages the complementary strengths of two distinct modes of reasoning: Chain-of-Thought (CoT) and Program-of-Thought (PoT). We describe a general framework for combining these two forms of reasoning in self-consistency, as well as particular strategies for both full sampling and early-stopping. We show that CoT-PoT ensembling not only improves overall accuracy, but also drastically reduces the number of samples required in comparison with the most efficient SC method. In particular, the majority of tasks can be addressed with *only two* samples, which has not been possible with any prior SC methods.

## 1 INTRODUCTION

Self-consistency (SC) is a widely adopted paradigm for improving reasoning in large language models (LLMs). In this ensembling approach, multiple outputs representing different reasoning paths are generated by the same model for a given input problem and the final answer is selected as the most frequently occurring one among these samples (Wang et al., 2022). While this approach yields higher accuracy than a single inference, it also comes with significantly higher computational cost, as numerous inference calls to the LLM are required, e.g. 40 samples are common to reach the best accuracy levels. Various approaches have therefore been proposed to reduce the number of samples required in self-consistency (Aggarwal et al., 2023; Li et al., 2024; Wang et al., 2025). For instance, *adaptive consistency* is an early-stopping technique where sampling is terminated if a confident majority is established early on (Aggarwal et al., 2023). While showing relative improvements in efficiency, such approaches still require many samples and yield at best comparable—and often lower—accuracy than full sampling. Hence, improving both the accuracy and efficiency of self-consistency is an important challenge, especially as inference-time scale-up is increasingly used to handle complex reasoning tasks with ever-larger models (DeepSeek-AI et al., 2025; OpenAI, 2024).

In this work we address this challenge with a novel approach to self-consistency that combines different reasoning modalities. The core idea behind self-consistency is that if different ways of reasoning converge on the same final answer, then such consensus serves as a strong signal of correctness. Hence, what matters is the *diversity* of the reasoning paths rather than just quantity. Existing SC techniques rely on high model temperatures to induce such diversity, but in practice we observe that this often yields reasoning traces that are very similar and may only have superficial syntactic variations in wording rather than substantial semantic differences. We address this issue with a new SC approach that is based on two fundamentally distinct modes of reasoning: chain-of-thought (*CoT*) (Wei et al., 2022) and program-of-thought (*PoT*) (Chen et al., 2023; Gao et al., 2023). CoT is a concrete form of reasoning in natural language where the model generates step-by-step inferences to explicitly construct a final answer. In contrast, PoT is a more abstract or symbolic form of reasoning where the model formulates the solution as a program that is executed to compute the final answer.

Figure 1 illustrates this contrast with an example query about bus scheduling along with a sample CoT and PoT solution (right). The CoT solution first makes a simplifying inference that the bus always stops at the same number of minutes past the hour, and then does all arithmetic calculations explicitly to infer the requested waiting time. Many other CoT samples from temperature sampling may follow a similar pattern of reasoning with variations only in style or phrasing. In contrast, the

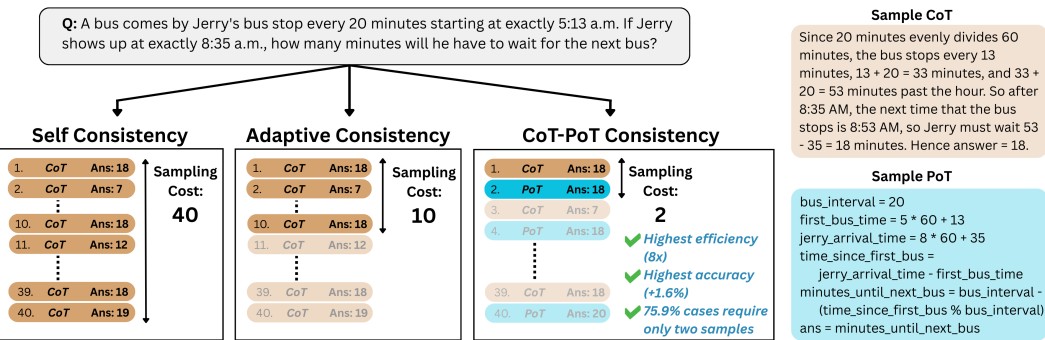

Figure 1: CoT-PoT consistency provides the highest accuracy, the highest efficiency and can solve most problems with only two samples, unlike any prior self-consistency method.

PoT solution creates a symbolic representation of the whole scenario, representing actual times as minutes past midnight, formulating the answer as the total time since the first bus modulo the interval, and using program execution to perform all the arithmetic calculations. Due to the different reasoning approaches, the two modalities also exhibit different kinds of errors: for instance, CoTs may perform calculation errors (e.g. "53 - 35 = 8"), while PoTs may incorrectly express symbolic relationships (e.g. use a division "/" operator instead of modulo "%"). Thus, the two modalities exhibit complementary strengths but have different error modes – CoT is more concrete and flexible but it can suffer from imprecision and computational errors, while PoT provides stronger computational robustness but may make symbolic formulation errors. The agreement between the two modalities would therefore mean alignment between logical framing and computational aspects.

Our *CoT-PoT ensembling* approach leverages the high diversity and complementary strengths of these two reasoning modalities to improve both the accuracy and efficiency of SC inference. Firstly, we explore cross-modal full sampling strategies that utilize the entire sampling budget, and show that aggregating over both CoT and PoT samples provides higher accuracy than either CoT-only or PoT-only self-consistency. However, another key observation of this work is that agreement between CoT and PoT responses also provides a strong signal for *early stopping* of sampling. Such agreement is informative since the error modes of the two approaches have very low correlation, unlike near-identical samples in the same modality that may make similar errors. We formalize this notion by first formulating a general Bayesian model of cross-modal agreement, and then study specialized instantiations of this model that represent different early-stopping sampling strategies. We investigate both data-driven strategies, in which model parameters are learned from held-out data, and data-independent strategies based on extreme parameterizations.

Figure 1 illustrates our most efficient early-stopping CoT-PoT method, where we alternately sample one CoT and one PoT solution until there is agreement between the two modalities. In this example scenario, we see that while the standard CoT-only self-consistency method requires the full budget of 40 samples, and the adaptive consistency early stopping method requires 10 samples to establish a statistical majority, our CoT-PoT method requires only 2 samples based on agreement between the initial CoT and PoT solutions while providing higher accuracy. Over a range of diverse benchmarks and different LLMs, we show how our early-stopping CoT-PoT techniques achieve higher accuracy than any prior self-consistency approaches, while also providing the highest reduction in the number of samples, by a factor of $8\times$. Furthermore, another distinguishing aspect of our approach is that most tasks can be handled with *only two* samples, which has not been possible with any prior self-consistency technique (the best early-stopping methods require at least four samples). On average, our CoT-PoT method can handle $75.9\%$ of tasks using only two samples, with this number exceeding $90\%$ for some benchmarks and models. This makes our method highly efficient for most problems, while still providing greater overall robustness over prior approaches.

In summary, we make the following contributions in this work: (1) We investigate full sampling methods combining CoT and PoT modalities, and show that such methods provide higher accuracy than standard SC under the same sampling budget. (2) We develop a general Bayesian model of cross-modal agreement, and derive both data-driven and data-independent early stopping sampling strategies as instantiations of this model. (3) We provide an empirical evaluation over five diverse

benchmarks and four mainstream LLMs (as well as one small model). This shows how CoT-PoT ensembling provides the highest accuracy and efficiency over prior methods, reducing number samples by 8× and solving most problems with only two samples.

## 2 CoT-PoT Ensembling

In this section we describe our cross-modal ensembling approach that combines CoT and PoT reasoning modalities to improve both the accuracy and efficiency of self-consistency. We explore such hybrid sampling strategies in two main directions: 1) *Full sampling* approaches that use the entire available decoding budget to maximize ensemble accuracy, and 2) *Early stopping* strategies that aim to minimize cost by dynamically terminating sampling once sufficient agreement between modalities is observed.

### 2.1 Full sampling CoT-PoT strategies

While standard self-consistency samples multiple CoT solutions and performs majority voting, in our hybrid approach we sample an equal number of CoT and PoT solutions for the full budget that is available. However, there can be multiple ways of aggregating the results across the different modalities to obtain the final answer. We consider various approaches to benefit from the complementary strengths of the two reasoning modalities.

Let $S^c$ and $S^p$ be the sequence of answers generated by CoT and PoT sampling for the same reasoning task, respectively. For any specific final answer $y$, let $F_c(y)$ and $F_p(y)$ be the number of instances that reach the final answer $y$ in $S^c$ and $S^p$, respectively. We consider the following combination strategies:

**CP$_{Maj}$ (Majority voting).** This approach performs a majority vote over all sampled answers from CoT and PoT combined. It leverages the overall diversity in the samples introduced by both modalities and simply takes the most frequently occurring answer. The result is obtained as:

$$y^* = \operatorname*{argmax}_y \left( F_c(y) + F_p(y) \right)$$

**CP$_{Max}$ (Maximum confidence modality).** In this approach we select the most frequent answer from either the CoT or PoT modality, whichever has higher frequency. This effectively lets the more confident modality dominate for each question. Formally, it is defined as:

$$y^* = \operatorname*{argmax}_y \left( \max \left( F_c(y), F_p(y) \right) \right)$$

**CP$_{Agr}$ (Modality agreement prioritization).** This approach first prioritizes answers that appear in both CoT and PoT, and then performs majority voting. Thus it gives the highest confidence to answers that see agreement between the two modalities. Formally, it is defined with a lexical ordering of two components: the indicator function for agreement and the total frequency for majority tie-breaking:

$$y^* = \operatorname*{argmax}_y \Big( \mathbf{1}\{y \in S^c \cap S^p\}, \; F_c(y) + F_p(y) \Big)_{\text{lex}}$$

### 2.2 Early stopping CoT-PoT strategies

While the full sampling strategies explore how accuracy may be maximized using all the available sampling budget, another important goal of this work is to leverage CoT-PoT agreement to minimize the number of samples and improve the efficiency of self-consistency techniques. Early stopping introduces a higher-stakes decision: we must decide whether to terminate sampling based on partial evidence without having seen the complete sample set. Hence, the main question is how we can measure such confidence based on agreement between the two modalities to determine when to stop. We investigate this question with a general Bayesian formulation based on the core agreement events of interest. We then describe concrete strategies that instantiate this general model, considering both *data-driven* strategies that infer seed probabilities from data, and simplified *data-independent* strategies based on edge-case parameterizations of the general model.

### 2.2.1 BAYESIAN AGREEMENT MODEL

We have a uniform sampling scheme that alternates between generating one CoT and one PoT answer until the total sampling budget is reached. During this iterative sampling, we aim to halt early whenever there is sufficiently strong confidence based on agreement between the two reasoning modalities. We model this process with parallel Bayesian hypothesis tests that continually update as sampling proceeds. Each test is anchored to a distinct answer generated as sampling progresses. Without loss of generality, let $y$ be a unique anchor answer generated by PoT at some iteration (the CoT case is treated symmetrically). This initiates a new hypothesis test that tracks the number of CoT agreements with $y$. For this or any subsequent iteration, let $t$ be the total number of CoT answers generated so far (whether before or after $y$), and let $k \leq t$ be the number of CoT answers that match $y$. Thus, formulated as a Bernoulli process with $t$ total trials and $k$ is the number of successes, each trial $i$ is defined as $A_i = 1$ if and only if $S^c[i] = y$.

The hypothesis test for $y$ is based on the event $C$ that the answer $y$ is *safe*: either $y$ is correct *or* the answer obtained from the full sampling will also be wrong. We define this as our target event of interest since we are aiming to infer high confidence for when to stop sampling, rather than strictly when the answer is correct (for which we may very rarely have very high confidence). Based on these events, the three base probabilities of interest are:

$$
\begin{array}{rcll}
c & = & P(C) & y \text{ is safe} \\
a_1 & = & P(A_i = 1) & \text{any CoT answer equals } y \\
a_2 & = & P(C \mid A_i = 1) & y \text{ is safe given agreement}
\end{array}
$$

From these we derive the likelihood of observing an agreement at trial $i$ under both hypotheses of $y$ being safe ($C$) or not ($\neg C$):

$$
q_1 = P(A_i = 1 \mid C) = \frac{a_1 a_2}{c} \qquad q_0 = P(A_i = 1 \mid \neg C) = \frac{a_1(1-a_2)}{1-c}
$$

Assuming independence between trials, the likelihood of observing $k$ successes in $t$ Bernoulli trials under each hypothesis is:

$$
P(k, t \mid C) = \binom{t}{k} q_1^k (1 - q_1)^{t-k} \qquad P(k, t \mid \neg C) = \binom{t}{k} q_0^k (1 - q_0)^{t-k}
$$

Finally, Bayes' rule provides the posterior probability that $y$ is safe after $k$ agreements in $t$ trials:

$$
P(C \mid k, t) = \frac{P(C)\, P(k,t \mid C)}{P(C)\, P(k,t \mid C) + P(\neg C)\, P(k,t \mid \neg C)} = \frac{c\, q_1^k (1-q_1)^{t-k}}{c\, q_1^k (1-q_1)^{t-k} + (1-c)\, q_0^k (1-q_0)^{t-k}}
$$

We determine an early stop as soon this posterior probability surpasses a desired confidence threshold $P(C \mid k, t) \geq \rho$. We explore two kinds of instantiation strategies of the general model above: data-driven variants that estimate model parameters from held-out agreement statistics, and heuristic variants that set extreme parameter values that reduce to simple stopping rules. In all cases the underlying sampling process samples one CoT and one PoT answer alternately, where the first CoT and PoT are generated at temperature 0 (as the LLM's most confident guess for each modality) and the rest at temperature 0.7 (for higher diversity (Wang et al., 2022)).

### 2.2.2 DATA-DRIVEN SPECIALIZATIONS

Our general Bayesian model is parameterized by the three core probabilities $c$, $a_1$ and $a_2$. These can be inferred statistically from data by performing full sampling and recording the rates of safety and agreement. For our data-driven strategies we infer these probabilities from the unused training sets of the benchmarks we use in our evaluation. With these inferred probabilities, we consider two specialized strategies based on which anchors are chosen to test agreements.

**CP$_{\text{DAA}}$** This is the *any-to-any* method where a new tracker is instantiated for every unique answer generated during sampling. Each tracker independently accumulates cross-modal matches, and we stop when any tracker's posterior probability exceeds the threshold.

**CP$_{DFA}$** This is a *first-to-any* approach where we create trackers only for the two initial, temperature-0 CoT and PoT answers ($S^c[0]$ and $S^p[0]$). We stop as soon as either initial answer establishes agreement from the other modality. This a more conservative strategy that only anchors on the highest confidence answers from the LLM.

### 2.2.3 DATA-INDEPENDENT SPECIALIZATIONS

Empirically, a key observation in this work is that the probability of safety given agreement is generally extremely high in practice, that is, $a_2 \approx 1$. If we consider the extreme case where $a_2 = 1$, this amounts to a strategy where we stop as soon as one cross-modal agreement is seen (the posterior is always 1 when $k = 1$ for any $t$). Based on this notion, we consider the following data-independent strategies for early-stopping.

**CP$_{AA}$** This is the any-to-any approach, where we stop as soon as there is agreement between any PoT and CoT answer. This is equivalent to **CP$_{DAA}$** when $a_2 = 1$.

**CP$_{FA}$** This is the first-to-any approach, where we stop as soon as there is any cross-modal agreement with the first PoT or CoT answer. This is equivalent to **CP$_{DFA}$** when $a_2 = 1$.

**CP$_{FF}$** This is the most conservative first-to-first strategy, where we only test agreement between the initial temperature-0 PoT and CoT answers (thus assuming $a_2 = 1$ and $t = 1$ with only one trial).

### 2.2.4 INCORPORATING ADAPTIVE CONSISTENCY

Although cross-modal agreement is a very strong signal when it happens, it is also possible that such agreement is not observed even though one answer becomes overwhelmingly dominant for one of the modalities as sampling progresses (especially if the problem is particularly suited to a specific modality). To capture this complementary evidence, we incorporate the adaptive consistency approach (Aggarwal et al., 2023) in all of our early-stopping strategies. This is done by including another parallel hypothesis test that implements the Beta-stopping rule of Aggarwal et al. (2023) in our alternating CoT-PoT sampling process. Let $v_1$ and $v_2$ be the current vote counts of the most frequent and the second-most frequent answers, respectively, aggregated over *all* CoT and PoT samples observed so far. Assuming a uniform $\text{Beta}(1, 1)$ prior on the true share $\theta$ of the leading answer, the posterior is modelled as $\text{Beta}(v_1 + 1, \ v_2 + 1)$, and the probability that the leader will remain the majority after unlimited additional sampling is simply the tail mass of this posterior Beta distribution above 0.5. We perform this Beta majority test in parallel with every cross-stream tracker and terminate as soon as any of these tests exceed the confidence threshold.

## 3 EVALUATION

In this section we present an evaluation of both the accuracy and efficiency of our full-sampling and early-stopping CoT-PoT ensembling methods.

***Datasets.*** We use five benchmarks covering a range of different kinds of reasoning tasks: **GSM8K** (Cobbe et al., 2021) consists of elementary to middle school level word problems; **MATH** (Hendrycks et al., 2021) consists of challenging high-school level math competition problems covering advanced topics including algebra, calculus and geometry; **FinQA** (Chen et al., 2021) contains problems from real-world financial contexts, requiring integrated reasoning over textual and structured data; **SVAMP** (Patel et al., 2021) contains arithmetic word problems designed to identify common numerical reasoning pitfalls in NLP models; and **TabMWP** (Lu et al., 2022) contains semi-structured problems involving reasoning with text and tabular data. For our evaluation we use 500 cases from the test splits of each dataset (to cap costs of our large sampling experiments).

***Models.*** We evaluate our methods over four different mainstream large language models: **GPT-3.5-Turbo** (OpenAI, 2022) and the more powerful **GPT-4-Omni** (OpenAI, 2024) models from OpenAI; **Mistral-large** (Mistral AI, 2024), which is a 123B parameter competitive reasoning model from Mistral; and the **Qwen3-Coder** 30B open source model with state-of-the-art coding capabilities (Yang et al., 2025).

***Sampling parameters.*** In our sampling process we use a maximum of 40 samples, as in prior work (Wang et al., 2022; Aggarwal et al., 2023). For CoT-only or PoT-only baseline methods, all of these

are either CoT samples or PoT samples. For our CoT-PoT methods, we sample one CoT and one PoT response alternately until the maximum budget. For all methods including baselines, the first samples for each modality are taken at temperature 0 and the rest at 0.7. Our CoT and PoT prompts are shown in the appendix. For uniformity, we use the same few shot examples for both CoT and PoT, with PoT prompts generally requiring fewer tokens. Finally, we use a confidence threshold of $\rho = 0.975$ for our early stopping CoT-PoT hypothesis tests.

***Seed probabilities inference.*** As discussed in Section 2.2.2, for our data-driven early stopping strategies we use held-out data to infer the three parameter probabilities of our Bayesian model: $c$ (safety), $a_1$ (agreement) and $a_2$ (safety given agreement). We randomly sample 100 problems from the training split of each of our datasets, perform full CoT-PoT ensembling, and compute maximum-likelihood estimates for each of the agreement

| Model | $c$ | $a_1$ | $a_2$ |
|---|---|---|---|
| GPT-3.5 | 0.784 | 0.505 | 0.997 |
| GPT-4O | 0.743 | 0.625 | 0.998 |
| MISTRAL | 0.784 | 0.671 | 0.992 |
| QWENCDR | 0.936 | 0.817 | 0.993 |

Table 1: Inferred parameter probabilities

events. These are used as the priors in all our data-driven strategies. Table 1 shows the inferred average probabilities for each LLM. In particular, we note the consistently high values for $a_2 \approx 1$, which illustrates the strong empirical correlation between cross-modal agreement and answer safety.

## 3.1 FULL SAMPLING RESULTS

The results of our full sampling CoT-PoT strategies are shown in Table 2. This shows the accuracy of each of our full sampling methods defined in Section 2.1. We compare these against the standard self-consistency approach (Wang et al., 2022), with the baseline methods $\mathsf{SC_{CoT}}$ and $\mathsf{SC_{PoT}}$ that perform a majority vote on CoT-only and PoT-only samples respectively. We also include for comparison the direct **CoT** and **PoT** baselines that represent the single temperature-0 sample from each modality. The table shows the results for each dataset (averaged across models), for each model (averaged across all datasets), as well as overall average.

| Dataset | $\mathsf{SC_{CoT}}$ | $\mathsf{SC_{PoT}}$ | $\mathsf{CP_{Maj}}$ | $\mathsf{CP_{Max}}$ | $\mathsf{CP_{Agr}}$ | CoT | PoT |
|---|---|---|---|---|---|---|---|
| GSM8K | 95.1 | 92.9 | **95.6** | 95.5 | 95.5 | 92.0 | 90.4 |
| MATH | 72.5 | 64.2 | **74.8** | **74.8** | 74.5 | 63.6 | 51.2 |
| SVAMP | 94.6 | 94.3 | **95.5** | 95.3 | **95.5** | 91.8 | 92.9 |
| FINQA | 60.7 | 62.9 | **62.6** | **62.6** | **62.6** | 57.8 | 61.1 |
| TABMWP | 79.7 | 85.4 | 84.3 | 84.3 | **84.8** | 78.5 | 82.5 |
| **Model** | | | | | | | |
| GPT-3.5 | 73.8 | 69.1 | **75.7** | 75.3 | 75.4 | 66.0 | 64.6 |
| GPT-4O | 83.5 | 83.1 | 85.1 | 85.0 | **85.2** | 80.8 | 77.8 |
| MISTRAL | 81.2 | 82.4 | 83.5 | **83.6** | **83.6** | 78.9 | 78.2 |
| QWENCDR | 83.5 | 85.1 | 85.9 | 85.9 | **86.1** | 81.2 | 81.7 |
| Average | 80.5 | 79.9 | **82.6** | 82.5 | **82.6** | 76.7 | 75.6 |

Table 2: Accuracy (%) of full sampling methods

The main result is that *CoT-PoT ensembling has higher accuracy than both CoT-only and PoT-only self-consistency*. All the CoT-PoT methods perform better than $\mathsf{SC_{CoT}}$ and $\mathsf{SC_{PoT}}$, with an overall average accuracy increase of 2.1%. Moreover, while CoT-only and PoT-only methods outperform each other on specific models and datasets, the CoT-PoT methods consistently perform better than both baselines on each of the five datasets and four language models. Although we do not observe a significant difference between the three CoT-PoT aggregation strategies, direct majority voting ($\mathsf{CP_{Maj}}$) and inter-modality agreement ($\mathsf{CP_{Agr}}$) perform slightly better than maximization within modalities ($\mathsf{CP_{Max}}$). This indicates that consensus and agreement between modalities provide additional benefits as opposed to competition between them.

## 3.2 EARLY-STOPPING SAMPLING RESULTS

Table 3 shows both the accuracy and efficiency (number of samples required) for each of our early-stopping CoT-PoT methods defined in Sections 2.2.2 and 2.2.3. We compare our methods against the two prior state-of-the-art early stopping approaches that use CoT-only sampling: **ASC** is the adaptive consistency method based on early majorities (Aggarwal et al., 2023), and **ESC** is the early-stopping approach based on sampling windows (Li et al., 2024). We make the following key observations:

1. *Every CoT-PoT method provides higher accuracy and efficiency than all prior early-stopping and full-sampling methods consistently across all datasets and models*. Our most efficient early-stopping method is $\mathsf{CP_{AA}}$. While having 1.6% higher accuracy than full sampling $\mathsf{SC_{CoT}}$ and both early-stopping

| | Accuracy | | | | | | | Number of samples | | | | | | |
|---|---|---|---|---|---|---|---|---|---|---|---|---|---|---|
| Dataset | ASC | ESC | $CP_{AA}$ | $CP_{FA}$ | $CP_{FF}$ | $CP_{DAA}$ | $CP_{DFA}$ | ASC | ESC | $CP_{AA}$ | $CP_{FA}$ | $CP_{FF}$ | $CP_{DAA}$ | $CP_{DFA}$ |
| GSM8K | 95.1 | 95.2 | 95.2 | 95.4 | **95.8** | 95.4 | 95.3 | 5.6 | 7.7 | **2.7** | 3.0 | 4.1 | 2.9 | 3.5 |
| MATH | 72.4 | 72.2 | 74.0 | 73.9 | **74.8** | 74.3 | 74.3 | 14.0 | 18.7 | **8.9** | 10.9 | 14.4 | 9.1 | 11.9 |
| SVAMP | 94.6 | 94.5 | 94.7 | 95.0 | **95.4** | 94.7 | 95.3 | 5.7 | 7.9 | **2.6** | 2.8 | 3.5 | **2.6** | 3.0 |
| FINQA | 60.6 | 60.7 | 62.2 | **62.4** | 62.3 | 62.2 | 62.3 | 8.5 | 11.9 | **4.4** | 4.8 | 7.6 | 4.5 | 6.1 |
| TABMWP | 79.8 | 79.7 | **84.7** | **84.7** | 84.2 | **84.7** | 84.5 | 6.8 | 8.9 | **6.5** | 6.8 | 8.5 | **6.5** | 7.0 |
| **Model** | | | | | | | | | | | | | | |
| GPT-3.5 | 73.8 | 73.8 | 74.7 | 75.1 | **75.7** | 75.0 | 75.3 | 12.4 | 17.1 | **6.3** | 7.8 | 11.6 | 6.5 | 8.9 |
| GPT-4O | 83.4 | 83.3 | 84.8 | 84.7 | **85.0** | 84.8 | 84.6 | 6.8 | 9.1 | **4.3** | 4.7 | 6.0 | **4.3** | 5.0 |
| MISTRAL | 81.2 | 81.1 | 83.0 | 83.1 | 83.3 | 83.2 | **83.5** | 6.9 | 9.2 | **5.0** | 5.5 | 6.8 | 5.1 | 5.9 |
| QWENCDR | 83.5 | 83.5 | 86.0 | **86.1** | 85.9 | 86.0 | 85.9 | 6.4 | 8.7 | **4.5** | 4.7 | 6.1 | **4.5** | 5.3 |
| **Average** | 80.5 | 80.4 | 82.1 | 82.3 | **82.5** | 82.3 | 82.3 | 8.1 | 11.0 | **5.0** | 5.7 | 7.6 | 5.1 | 6.3 |
| $\Delta$-CP$_{Maj}$ | -2.1 | -2.2 | -0.5 | -0.3 | **-0.1** | -0.3 | -0.3 | 4.9x | 3.6x | **8.0x** | 7.0x | 5.3x | 7.8x | 6.3x |
| $\Delta$-SC$_{CoT}$ | 0.0 | -0.1 | +1.6 | +1.8 | **+2.0** | +1.8 | +1.8 | 4.9x | 3.6x | **8.0x** | 7.0x | 5.3x | 7.8x | 6.3x |

Table 3: Accuracy and number of samples for early-stopping strategies. $\Delta$ rows show the difference in accuracy and factor of reduction in the number of samples in comparison to full-sampling methods.

baselines, this method also provides the biggest efficiency improvement among all methods. Overall, it reduces the number of samples drastically by a factor of $8\times$ as compared to $4.9\times$ by the best prior early-stopping method **ASC**. It also consistently shows the best efficiency for each of the datasets and models. On the other hand, our most accurate early-stopping method is the conservative approach of **CP$_{FF}$**, which has $2\%$ higher accuracy than all prior early-stopping and full sampling methods, and only a $0.1\%$ accuracy drop compared to the most accurate full sampling method, which is our **CP$_{Maj}$**. It also provides better efficiency than prior early-stopping methods: $5.3\times$ compared to $4.9\times$ by **ASC** and $3.6\times$ by **ESC**.

2. *Early-stopping with CoT-PoT requires only two samples in the majority of cases*. Figure 2 shows the percentage of test cases that can be solved with just two samples (one CoT and one PoT) by our CoT-PoT methods (this percentage is the same for all the CoT-PoT early-stopping methods). Overall, on average across all models and datasets, CoT-PoT methods can terminate with just 2 samples in $75.9\%$ of cases. No prior technique can terminate with only two samples in any scenario: **ASC** requires a minimum of 4 and **ESC** requires at least 5 samples in all cases.

We also observe from Table 3 that the data-driven parameter estimation methods (**CP$_{DAA}$** and **CP$_{DFA}$**) do not yield the most optimal results, though they are generally more accurate and less efficient than their data-independent counterparts. This can be expected as our two best methods **CP$_{AA}$** and **CP$_{FF}$** respectively model the most aggressive and conservative extremes of our CoT-PoT approach, and these simple data-independent strategies show how our approach can be applied easily without the need for pa-

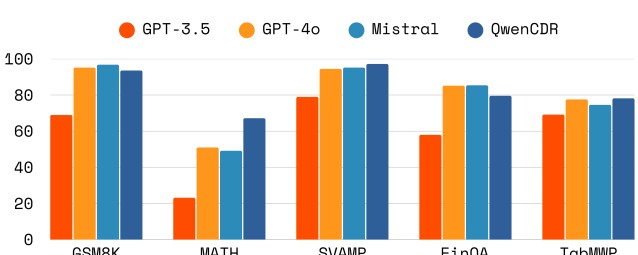

Figure 2: Percentage of problems solved with only two samples by early-stopping CoT-PoT methods.

rameter estimation from data. Empirically, we find the extremely high values of $a_2$ (safety given agreement probability) diminish the need for parameter estimation from data over these benchmarks and models (as even a single cross-modal agreement provides a strong stopping signal). However, in other scenarios where cross-modal safety may be lower, the data-driven methods may provide higher robustness. We describe one such scenario for a small 3B parameter model in Section 3.4.

### 3.3 ABLATION STUDIES

We performed ablation experiments to further examine different aspects of our cross-modal approach. Firstly, we evaluated the benefit of early stopping using cross-modal agreement versus only using adaptive consistency over a mix of CoT and PoT samples. We test this with the $A_{ASC-CP}$ ablation, which is adaptive consistency applied over our hybrid sampling scheme of alternating CoT and PoT samples (rather than CoT-only samples as in $ASC$). Table 4 shows the ablation results: while $A_{ASC-CP}$ reaches the same best accuracy as $CP_{FF}$, it requires more samples (9.7 vs 7.6). This shows how cross-modal agreement provides further efficiency gains over adaptive consistency without loss of accuracy.

Another aspect we examined is how cross-modal agreement compares with intra-modal agreement. We define the ablation $A_{FS-C}$ as standard CoT-only self-consistency but where an answer is returned early if there is agreement between the first and second CoT samples. $A_{FS-P}$ is defined similarly but

|  | $CP_{Maj}$ | $CP_{AA}$ | $CP_{FF}$ | $A_{ASC-CP}$ | $A_{FS-C}$ | $A_{FS-P}$ |
|---|---|---|---|---|---|---|
| **Accuracy** | 82.6 | 82.1 | 82.5 | 82.5 | 79.9 | 78.1 |
| **#Samples** | 40.0 | 5.0 | 7.6 | 9.7 | 7.5 | 6.5 |

Table 4: Ablations compared to CoT-PoT methods

with PoT-only samples. These methods are in contrast to $CP_{FF}$ which stops on agreement between the first CoT and PoT samples. The results in Table 4 show a significant drop in accuracy for both these methods, not only relative to $CP_{FF}$, but also to their respective full sampling counterparts $SC_{CoT}$ and $SC_{PoT}$ (accuracies shown in Table 2). This indicates the robustness of cross-modal agreement as an early stopping signal in comparison to intra-modal agreement for either modality.

We also investigated how efficiency of early-stopping methods changes with sampling budget. Figure 3 shows the number of samples used as the maximum sampling budget increases from 10 to 40. While all methods utilize more samples as the budget increases, we observe a greater gap between the baselines and our most efficient method $CP_{AA}$ as budget increases. This shows how the efficiency gains with CoT-PoT ensembling increase with higher sampling budgets.

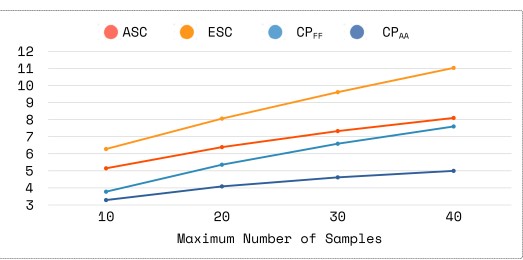

Figure 3: Efficiency vs. sampling budget

### 3.4 CASE STUDY: INDUCING POT FROM COT

While large commercial-scale models generally exhibit CoT and PoT modalities well as seen in our main results, smaller open models may often lack strong PoT capabilities. Moreover, PoT training data may not be readily available, and prior work often relies on expensive teacher models like GPT-4 to generate PoT rationales (Yue et al., 2024b; Gou et al., 2024). To investigate this limitation, we conducted a case study on a small 3B-parameter Llama 3.2 model on the most challenging MATH benchmark. PoT was indeed significantly weaker than CoT in this model (18.8% vs. 35.2%), though CoT-PoT still outperformed standard SC. To improve PoT performance, we explored a *bootstrapping* approach where PoT reasoning was self-induced from CoT: using questions and the CoT data from 4000 problems from the train set, we prompted the base model to generate PoT rationales (prompt in Appendix). Only outputs matching the ground truth were retained, and we iteratively fine-tuned on these to generate further PoT rationales, yielding 2790 in total. This is similar to self-training methods (Zelikman et al., 2022), except that we inferred one reasoning modality from another. Finally, we used all of the self-generated PoT data to fine-tune a lightweight LoRA adapter for PoT on the base model and a similar CoT adapter using the original CoT data. This enabled a simple switch between modalities: during ensembling, the relevant adapter was activated when sampling from each modality.

Table 5 presents results for both the base model and the CoT-PoT enhanced models on the MATH test set. Firstly, even on the weak base model, the CoT-PoT methods outperformed prior methods in both accuracy and efficiency. With the LoRA adapters enabled, there was significant PoT improvement, which led to bigger accuracy improvements across all CoT-PoT ensembling strategies over CoT-only ensembling (even though there was less bootstrapped data for PoT training than for CoT). However, we notice that in this case $CP_{AA}$ had lower accuracy than $SC_{CoT}$ (49.0 vs. 49.8), but its data-driven counterpart $CP_{DAA}$ is the most efficient method that has higher than baseline accuracy (51.0). This shows the robustness gain from data-driven inference in this weaker model setting. Overall, this study

| Model | Accuracy | | | | | | | | Number of samples | | | |
|---|---|---|---|---|---|---|---|---|---|---|---|---|
| | CoT | PoT | $SC_{CoT}$ | $CP_{Maj}$ | ASC | $CP_{AA}$ | $CP_{FF}$ | $CP_{DAA}$ | ASC | $CP_{AA}$ | $CP_{FF}$ | $CP_{DAA}$ |
| LLAMA3B | 35.2 | 18.8 | 46.0 | **47.8** | 45.8 | 47.8 | 47.8 | 47.8 | 13.7 | 10.6 | 15.0 | 10.6 |
| LLAMA3B-CP | 36.4 | 32.4 | 49.8 | **52.6** | 49.8 | 49.0 | 52.2 | 51.0 | 14.1 | 8.3 | 13.6 | 10.9 |

Table 5: Accuracy and efficiency before and after SFT with PoT self-induction on Llama 3B model

illustrates an interesting research direction where one reasoning modality can be effectively improved by another, and still provide overall cross-modal ensembling benefits.

## 4 RELATED WORK

**Test Time Scaling for LLMs.** Test-time scaling strategies fall into two main categories: sequential refinement and parallel sampling. Sequential methods, such as long chains of thought OpenAI (2024); Guo et al. (2025) and self-correction (Huang et al., 2022; Madaan et al., 2024; Lee et al., 2025), guide models through multi-step reasoning and revision. While widely adopted in recent models, much of their development focuses on training-time integration. In contrast, parallel approaches like Best-of-$N$ improve solution coverage by generating multiple responses (Chollet, 2019; Irvine et al., 2023; Brown et al., 2024), though selecting the correct solution remains challenging (Brown et al., 2024; Hassid et al., 2024; Christiano et al., 2017; Wang et al., 2024). In this context, self-consistency (SC) has emerged as an effective tool for test-time inference (Wang et al., 2022). In this work, we explore improving both the accuracy and efficiency of SC via cross-modal reasoning.

**Improving efficiency of self-consistency.** While SC improves accuracy, it comes with high inference-time cost, and various approaches have been proposed to improve its efficiency. Adaptive consistency uses early stopping when a confident majority emerges during sampling (Aggarwal et al., 2023). A related approach (Li et al., 2024) checks for early majority within small windows, though this always requires a fixed minimal sample size. (Wang et al., 2025) propose allocating sampling budgets based on problem difficulty, but this only works over large problem batches to infer relative difficulty. All of these methods operate within the single CoT modality and depend on majority or difficulty estimation. In contrast, we propose a cross-modal ensembling approach that leverages agreement between CoT and PoT reasoning as a strong early stopping signal. We have shown how our approach provides both higher accuracy and drastic efficiency gains over prior approaches, including inference from only two samples, which no prior method can provide.

**Combining Chain-of-Thought and Program-of-Thought.** Recent work has also investigated combinations of CoT and PoT reasoning in various ways. Yue et al. (2024b) show the value of fine-tuning models on both CoT and PoT data and using the two modalities for different kinds of problems. LLM cascades (Yue et al., 2024a) use large numbers of mixed samples of CoT and PoT rationales as a gating mechanism to decide whether a task should be solved by a smaller LLM or escalated to a larger model. Other approaches devise specialized LLM prompting algorithms that incorporate CoT and PoT in different ways, e.g. question generalization (Imani et al., 2023) or assigning different modalities to different kinds of problems (Liu et al., 2023). All of these approaches propose specialized techniques that leverage the two modalities in different ways. In contrast, we propose an incorporation of the two modalities within the general self-consistency paradigm and show how this not only provides higher accuracy, but also a drastic improvement in the efficiency of self-consistency, which no prior work has shown.

## 5 CONCLUSION

We have presented a cross-modal ensembling approach that combines Chain-of-Thought and Program-of-Thought reasoning to improve both the accuracy and efficiency of self-consistency in LLMs. Our experiments across diverse benchmarks and models show that CoT-PoT ensembling consistently outperforms standard CoT-only approaches. In particular, it provides highly efficient early stopping, often requiring just two samples—which has not been achievable with any prior technique.

**LLM use.** LLMs were used in this project to aid/polish paper writing, formalizing and checking mathematical formulations, and finding related research works.

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

# A APPENDIX

## A.1 PROMPTS

This section contains the full prompts we used for chain-of-thought and program-of-thought inference.

```
CoT Prompt - GSM

Please solve the given mathematical problem, doing step by step reasoning to arrive
at the final answer. Please mark the final answer in a "\\boxed{}" annotation as
shown in the example below.
------
Problem:
Let \\[f(x) = \\left\\{\n\\begin{array}{cl} ax+3, &\\text{ if }x>2, \\\\\nx-5
&\\text{ if } -2 \\le x \\le 2, \\\\\n2x-b &\\text{ if } x <-2.\n\\end{array}
\n\\right.\\]
Find $a+b$ if the piecewise function is continuous (which means
that its graph can be drawn without lifting your pencil from the paper).
Solution:
For the piecewise function to be continuous, the cases must "meet" at $2$ and $-2$.
For example, $ax+3$ and $x-5$ must be equal when $x=2$. This implies $a(2)+3=2-5$,
which we solve to get $2a=-6 \\Rightarrow a=-3$. Similarly, $x-5$ and $2x-b$ must
be equal when $x=-2$. Substituting, we get $-2-5=2(-2)-b$, which implies $b=3$.
So $a+b=-3+3=\\boxed{0}$.
------
Problem:
Jame's buys 100 head of cattle for $40,000.  It cost 20% more than that to feed them.
They each weigh 1000 pounds and sell for $2 per pound.
How much profit did he make?
Solution:
It cost .2*40000=$<<.2*40000=8000>>8000 more to feed than buy them
\nSo the total cost to feed them was 40000+8000=$<<40000+8000=48000>>48,000
\nSo in total it cost 48000+40000=$<<48000+40000=88000>>88,000
\nEach cow sells for 2*1000=$<<2*1000=2000>>2000\nSo he sells them for
2000*100=$<<2000*100=200000>>200,000
\nSo he makes a profit of 200,000-88,000=$<<200000-88000=112000>>112,000
\\\\boxed{112000}
------
Problem:
[[PROBLEM]]
```

**Figure 4**: CoT prompt used for GSM8K, MATH, and SVAMP (two in-domain demonstrations).

```
PoT Prompt - GSM

Please generate Python code to solve the given mathematical problem. The code should
store the final answer in a variable named 'ans' as shown in the example below.
------
Question:
Let \\[f(x) = \\left\\{\n\\begin{array}{cl} ax+3, &\\text{ if }x>2,
\\\\\nx-5 &\\text{ if } -2 \\le x \\le 2, \\\\\n2x-b &\\text{ if } x <-2.
\n\\end{array}\n\\right.\\]
Find $a+b$ if the piecewise function is continuous (which means that its graph can
be drawn without lifting your pencil from the paper).
PythonCode:
from sympy import symbols, Eq, solve

a, b = symbols('a b')
eqs = [
    Eq(2 * a + 3, -3),   # Match limits at x = 2
    Eq(-4 - b, -7)       # Match limits at x = -2
]
sol = solve(eqs, (a, b))
a_val, b_val = sol[a], sol[b]
ans = a_val + b_val
------
Question:
Jame's buys 100 head of cattle for $40,000.  It cost 20% more than that to feed them.
They each weigh 1000 pounds and sell for $2 per pound.  How much profit did he make?
PythonCode:
num_cattle = 100
purchase_price = 40_000
feed_cost = purchase_price * 1.20
weight_per_cow = 1_000
price_per_pound = 2
revenue = num_cattle * weight_per_cow * price_per_pound
total_cost = purchase_price + feed_cost
ans = revenue - total_cost
------
Question:
[[QUESTION]]
PythonCode:
```

**Figure 5**: PoT prompt used for GSM8K, MATH, and SVAMP (two in-domain demonstrations).

## CoT Prompt - FinQA

Please solve the given mathematical problem, doing step by step reasoning to arrive at
the final answer. Please mark the final answer in a "\\boxed{}" annotation. Be mindful of
units handling in your solution.
------
Problem:
for uncoated freesheet paper and market pulp announced at the end of 2009 become
effective . input costs are expected to be higher due to wood supply constraints at the kwidzyn
mill and annual tariff increases on energy in russia . planned main- tenance outage costs are
expected to be about flat , while operating costs should be favorable . asian printing papers net
sales were approx- imately $ 50 million in 2009 compared with approx- imately $ 20 million in
both 2008 and 2007 . operating earnings increased slightly in 2009 compared with 2008 , but
were less than $ 1 million in all periods . u.s . market pulp net sales in 2009 totaled $ 575
million compared with $ 750 million in 2008 and $ 655 million in 2007 . operating earnings in
2009 were $ 140 million ( a loss of $ 71 million excluding alter- native fuel mixture credits
and plant closure costs ) compared with a loss of $ 156 million ( a loss of $ 33 million
excluding costs associated with the perma- nent shutdown of the bastrop mill ) in 2008 and earn-
ings of $ 78 million in 2007 . sales volumes in 2009 decreased from 2008 levels due to weaker
global demand . average sales price realizations were significantly lower as the decline in
demand resulted in significant price declines for market pulp and smaller declines in fluff
pulp . input costs for wood , energy and chemicals decreased , and freight costs were
significantly lower . mill operating costs were favorable across all mills , and planned maintenance
downtime costs were lower . lack-of-order downtime in 2009 increased to approx- imately 540000
tons , including 480000 tons related to the permanent shutdown of our bastrop mill in the
fourth quarter of 2008 , compared with 135000 tons in 2008 . in the first quarter of 2010 ,
sales volumes are expected to increase slightly , reflecting improving customer demand for
fluff pulp , offset by slightly seasonally weaker demand for softwood and hard- wood pulp in
china . average sales price realizations are expected to improve , reflecting the realization
of previously announced sales price increases for fluff pulp , hardwood pulp and softwood
pulp . input costs are expected to increase for wood , energy and chemicals , and freight
costs may also increase . planned maintenance downtime costs will be higher , but operating
costs should be about flat . consumer packaging demand and pricing for consumer packaging
prod- ucts correlate closely with consumer spending and general economic activity . in
addition to prices and volumes , major factors affecting the profitability of consumer packaging
are raw material and energy costs , freight costs , manufacturing efficiency and product mix
. consumer packaging net sales in 2009 decreased 4% ( 4 % ) compared with 2008 and
increased 1% ( 1 % ) compared with 2007 . operating profits increased significantly compared with
both 2008 and 2007 . excluding alternative fuel mixture credits and facility closure costs ,
2009 operating profits were sig- nificantly higher than 2008 and 57% ( 57 % ) higher than
2007 . benefits from higher average sales price realizations ( $ 114 million ) , lower raw
material and energy costs ( $ 114 million ) , lower freight costs ( $ 21 million ) , lower costs
associated with the reorganiza- tion of the shorewood business ( $ 23 million ) , favor- able
foreign exchange effects ( $ 14 million ) and other items ( $ 12 million ) were partially offset
by lower sales volumes and increased lack-of-order downtime ( $ 145 million ) and costs
associated with the perma- nent shutdown of the franklin mill ( $ 67 million ) . additionally ,
operating profits in 2009 included $ 330 million of alternative fuel mixture credits . consumer
packaging in millions 2009 2008 2007 .
-----------------------------------------------------------
\n| in millions     | 2009  | 2008  | 2007  |
\n|-----------------|--------|--------|--------|
\n| sales           | $ 3060 | $ 3195 | $ 3015 |
\n| operating profit | 433   | 17    |112    |
\n-----------------------------------------------------------
sales: 2009: $3060, 2008: $ 3195, 2007: $ 3015.
operating profit: 2009: 433, 2008: 17, 2007: 112.
north american consumer packaging net sales were $ 2.2 billion compared with $ 2.5 billion in 2008 and
$ 2.4 billion in 2007 . operating earnings in 2009 were $ 343 million ( $ 87 million
excluding alter- native fuel mixture credits and facility closure costs ) compared with $ 8
million ( $ 38 million excluding facility closure costs ) in 2008 and $ 70 million in 2007 .
coated paperboard sales volumes were lower in 2009 compared with 2008 reflecting weaker market
conditions . average sales price realizations were significantly higher , reflecting the full-year
realization of price increases implemented in the second half of 2008 . raw material costs for wood
, energy and chemicals were significantly lower in 2009 , while freight costs were also
favorable . operating costs , however , were unfavorable and planned main- tenance downtime costs
were higher . lack-of-order downtime increased to 300000 tons in 2009 from 15000 tons in 2008
due to weak demand . operating results in 2009 include income of $ 330 million for
alternative fuel mixture credits and $ 67 million of expenses for shutdown costs for the franklin
mill . foodservice sales volumes were lower in 2009 than in 2008 due to generally weak

```
world-wide economic conditions . average sales price realizations were . what is the average value
for sales?

Solution:
To determine the average value for sales across the three years
provided (2009: $3060M, 2008: $3195M, 2007: $3015M), sum them and divide by 3: (3060 + 3195 +
3015) / 3 = 9270 / 3 = 3090. The final answer is \(\boxed{3090}\).
------
Problem:
interest rate cash flow hedges 2013 we report changes in the fair value of cash flow hedges in
accumulated other comprehensive loss until the hedged item affects earnings . at both december 31 ,
2008 and 2007 , we had reductions of $ 4 million recorded as an accumulated other
comprehensive loss that is being amortized on a straight-line basis through september 30 , 2014 . as
of december 31 , 2008 and 2007 , we had no interest rate cash flow hedges outstanding .
earnings impact 2013 our use of derivative financial instruments had the following impact on
pre-tax income for the years ended december 31 : millions of dollars 2008 2007 2006 .
---------------------------------------------------------------------------------
\n| millions of dollars                                        |2008 | 2007  | 2006    |
\n|--------------------------------------------------------------|----|--------|---------|
\n| ( increase ) /decrease in interest expense from interest rate hedging |$ 1  | $ -8 (8) | $ -8 ( 8 ) |
\n| ( increase ) /decrease in fuel expense from fuel derivatives    | 1  |-1 ( 1 ) | 3        |
\n| increase/ ( decrease ) in pre-tax income                      | $ 2| $ -9 (9) | $ -5 (5)|
\n--------------------------------------------------------------------------------
( increase ) /decrease in interest expense from interest rate hedging: 2008: $ 1, 2007:
$ -8 ( 8 ), 2006: $ -8 ( 8 ). ( increase ) /decrease in fuel expense from fuel
derivatives: 2008: 1, 2007: -1 ( 1 ), 2006: 3. increase/ ( decrease ) in pre-tax income: 2008: $ 2,
2007: $ -9 ( 9 ), 2006: $ -5 ( 5 ). fair value of debt instruments 2013 the fair value of our
short- and long-term debt was estimated using quoted market prices , where available , or
current borrowing rates . at december 31 , 2008 , the fair value of total debt is approximately
$ 247 million less than the carrying value . at december 31 , 2007 , the fair value of
total debt exceeded the carrying value by approximately $ 96 million . at december 31 , 2008
and 2007 , approximately $ 320 million and $ 181 million , respectively , of fixed-rate debt
securities contained call provisions that allowed us to retire the debt instruments prior to final
maturity , with the payment of fixed call premiums , or in certain cases , at par . sale of
receivables 2013 the railroad transfers most of its accounts receivable to union pacific receivables
, inc . ( upri ) , a bankruptcy-remote subsidiary , as part of a sale of receivables
facility . upri sells , without recourse on a 364-day revolving basis , an undivided interest in
such accounts receivable to investors . the total capacity to sell undivided interests to
investors under the facility was $ 700 million and $ 600 million at december 31 , 2008 and 2007 ,
respectively . the value of the outstanding undivided interest held by investors under the facility
was $ 584 million and $ 600 million at december 31 , 2008 and 2007 , respectively . upri
reduced the outstanding undivided interest held by investors due to a decrease in available
receivables at december 31 , 2008 . the value of the outstanding undivided interest held by
investors is not included in our consolidated financial statements . the value of the undivided
interest held by investors was supported by $ 1015 million and $ 1071 million of accounts
receivable held by upri at december 31 , 2008 and 2007 , respectively . at december 31 , 2008 and
2007 , the value of the interest retained by upri was $ 431 million and $ 471 million ,
respectively . this retained interest is included in accounts receivable in our consolidated
financial statements . the interest sold to investors is sold at carrying value , which
approximates fair value , and there is no gain or loss recognized from the transaction . the value of
the outstanding undivided interest held by investors could fluctuate based upon the
availability of eligible receivables and is directly affected by changing business volumes and credit
risks , including default and dilution . if default or dilution percentages were to increase
one percentage point , the amount of eligible receivables would decrease by $ 6 million .
should our credit rating fall below investment grade , the value of the outstanding undivided
interest held by investors would be reduced , and , in certain cases , the investors would have
the right to discontinue the facility . the railroad services the sold receivables ; however
, the railroad does not recognize any servicing asset or liability as the servicing fees
adequately compensate us for these responsibilities . the railroad collected approximately $ 17.8
billion and $ 16.1 billion during the years ended december 31 , 2008 and 2007 , respectively .
upri used certain of these proceeds to purchase new receivables under the facility. . what
was the difference in billions of sold receivables from 2007 to 2008?

Solution:
From the text, 2007 receivables sold were $16.1 billion while 2008's were $17.8 billion.
The difference is: 17.8 - 16.1 = 1.7. The final answer is \(\boxed{1.7}\).
------
Problem:
baker hughes ,age company notes to consolidated and combined financial statements bhge 2017 form 10-k |
```

```
83 issuance pursuant to awards granted under the lti plan over its term which expires on
the date of the annual meeting of the company in 2027 . a total of 53.7 million shares of
class a common stock are available for issuance as of december 31 , 2017 . as a result of the
acquisition of baker hughes , on july 3 , 2017 , each outstanding baker hughes stock option was
converted into an option to purchase a share of class a common stock in the company . consequently
, we issued 6.8 million stock options which are fully vested . each converted option is
subject to the same terms and conditions as applied to the original option , and the per share
exercise price of each converted option was reduced by $ 17.50 to reflect the per share amount of
the special dividend pursuant to the agreement associated with the transactions .
additionally , as a result of the acquisition of baker hughes , there were 1.7 million baker hughes
restricted stock units ( rsus ) that were converted to bhge rsus at a fair value of $ 40.18 .
stock-based compensation cost is measured at the date of grant based on the calculated fair value of
the award and is generally recognized on a straight-line basis over the vesting period of
the equity grant . the compensation cost is determined based on awards ultimately expected
to vest ; therefore , we have reduced the cost for estimated forfeitures based on
historical forfeiture rates . forfeitures are estimated at the time of grant and revised , if
necessary , in subsequent periods to reflect actual forfeitures . there were no stock-based
compensation costs capitalized as the amounts were not material . during the year ended december 31 ,
2017 , we issued 2.1 million rsus and 1.6 million stock options under the lti plan . these
rsus and stock options generally vest in equal amounts over a three-year vesting period
provided that the employee has remained continuously employed by the company through such vesting
date . stock based compensation expense was $ 37 million in 2017 . included in this amount is
$ 15 million of expense which relates to the acceleration of equity awards upon
termination of employment of baker hughes employees with change in control agreements , and are
included as part of \"merger and related costs\" in the consolidated and combined statements of
income ( loss ) . as bhge llc is a pass through entity , any tax benefit would be recognized by
its partners . due to its cumulative losses , bhge is unable to recognize a tax benefit on
its share of stock related expenses . stock options the fair value of each stock option
granted is estimated using the black-scholes option pricing model . the following table presents
the weighted average assumptions used in the option pricing model for options granted under
the lti plan . the expected life of the options represents the period of time the options
are expected to be outstanding . the expected life is based on a simple average of the
vesting term and original contractual term of the awards . the expected volatility is based on
the historical volatility of our five main competitors over a six year period . the
risk-free interest rate is based on the observed u.s . treasury yield curve in effect at the time
the options were granted . the dividend yield is based on a five year history of dividend
payouts in baker hughes. .
-----------------------------------------------------------------------
\n|                                               | 2017            |
\n|-----------------------------------------------|-----------------|
\n| expected life ( years )                       | 6               |
\n| risk-free interest rate                       | 2.1% ( 2.1 % )  |
\n| volatility                                    | 36.4% ( 36.4 % ) |
\n| dividend yield                                | 1.2% ( 1.2 % )  |
\n| weighted average fair value per share at grant date | $12.32    |
\n-----------------------------------------------------------------------
expected life ( years ): 2017: 6. risk-free interest rate: 2017: 2.1% ( 2.1 % ).
volatility: 2017: 36.4% ( 36.4 % ). dividend yield: 2017: 1.2% ( 1.2 % ). weighted average fair
value per share at grant date: 2017: $ 12.32. what is the total value of rsus converted to
bhge rsus , in millions?

Solution:
From the text, 1.7 million RSUs were converted at $40.18 each,
so 1.7 × 40.18 = 68.306 → approximately 68.3. The final answer is \(\boxed{68.3}\).
------
Problem:
for uncoated freesheet paper and market pulp announced at the end of 2009
become effective . input costs are expected to be higher due to wood supply constraints at the
kwidzyn mill and annual tariff increases on energy in russia . planned main- tenance outage
costs are expected to be about flat , while operating costs should be favorable . asian
printing papers net sales were approx- imately $ 50 million in 2009 compared with approx- imately
$ 20 million in both 2008 and 2007 . operating earnings increased slightly in 2009
```

```
compared with 2008 , but were less than $ 1 million in all periods . u.s . market pulp net sales
in 2009 totaled $ 575 million compared with $ 750 million in 2008 and $ 655 million in 2007
. operating earnings in 2009 were $ 140 million ( a loss of $ 71 million excluding alter-
native fuel mixture credits and plant closure costs ) compared with a loss of $ 156 million ( a
loss of $ 33 million excluding costs associated with the perma- nent shutdown of the bastrop
mill ) in 2008 and earn- ings of $ 78 million in 2007 . sales volumes in 2009 decreased from
2008 levels due to weaker global demand . average sales price realizations were significantly
lower as the decline in demand resulted in significant price declines for market pulp and
smaller declines in fluff pulp . input costs for wood , energy and chemicals decreased , and
freight costs were significantly lower . mill operating costs were favorable across all mills ,
and planned maintenance downtime costs were lower . lack-of-order downtime in 2009 increased
to approx- imately 540000 tons , including 480000 tons related to the permanent shutdown of
our bastrop mill in the fourth quarter of 2008 , compared with 135000 tons in 2008 . in the
first quarter of 2010 , sales volumes are expected to increase slightly , reflecting improving
customer demand for fluff pulp , offset by slightly seasonally weaker demand for softwood and
hard- wood pulp in china . average sales price realizations are expected to improve ,
reflecting the realization of previously announced sales price increases for fluff pulp , hardwood
pulp and softwood pulp . input costs are expected to increase for wood , energy and chemicals
, and freight costs may also increase . planned maintenance downtime costs will be higher
, but operating costs should be about flat . consumer packaging demand and pricing for
consumer packaging prod- ucts correlate closely with consumer spending and general economic
activity . in addition to prices and volumes , major factors affecting the profitability of
consumer packaging are raw material and energy costs , freight costs , manufacturing efficiency
and product mix . consumer packaging net sales in 2009 decreased 4% ( 4 % ) compared with
2008 and increased 1% ( 1 % ) compared with 2007 . operating profits increased significantly
compared with both 2008 and 2007 . excluding alternative fuel mixture credits and facility
closure costs , 2009 operating profits were sig- nificantly higher than 2008 and 57% ( 57 % )
higher than 2007 . benefits from higher average sales price realizations ( $ 114 million ) ,
lower raw material and energy costs ( $ 114 million ) , lower freight costs ( $ 21 million ) ,
lower costs associated with the reorganiza- tion of the shorewood business ( $ 23 million ) ,
favor- able foreign exchange effects ( $ 14 million ) and other items ( $ 12 million ) were
partially offset by lower sales volumes and increased lack-of-order downtime ( $ 145 million ) and
costs associated with the perma- nent shutdown of the franklin mill ( $ 67 million ) .
additionally , operating profits in 2009 included $ 330 million of alternative fuel mixture credits .
consumer packaging in millions 2009 2008 2007 .
-------------------------------------------------------------
\n| in millions    | 2009  | 2008   | 2007   |
\n|----------------|-------|--------|--------|
\n| sales          | $ 3060 | $ 3195 | $ 3015 |
\n| operating profit | 433 | 17     | 112    |
\n-------------------------------------------------------------
sales: 2009: $3060, 2008: $ 3195, 2007: $ 3015.
operating profit: 2009: 433, 2008: 17, 2007: 112.
north american consumer packaging net sales were $ 2.2 billion compared with $ 2.5 billion in 2008 and
$ 2.4 billion in 2007 . operating earnings in 2009 were $ 343 million ( $ 87 million
excluding alter- native fuel mixture credits and facility closure costs ) compared with $ 8
million ( $ 38 million excluding facility closure costs ) in 2008 and $ 70 million in 2007 .
coated paperboard sales volumes were lower in 2009 compared with 2008 reflecting weaker market
conditions . average sales price realizations were significantly higher , reflecting the full-year
realization of price increases implemented in the second half of 2008 . raw material costs for wood
, energy and chemicals were significantly lower in 2009 , while freight costs were also
favorable . operating costs , however , were unfavorable and planned main- tenance downtime costs
were higher . lack-of-order downtime increased to 300000 tons in 2009 from 15000 tons in 2008
due to weak demand . operating results in 2009 include income of $ 330 million for
alternative fuel mixture credits and $ 67 million of expenses for shutdown costs for the franklin
mill . foodservice sales volumes were lower in 2009 than in 2008 due to generally weak
world-wide economic conditions . average sales price realizations were . considering the years 2008
and 2009 , what is the variation observed in the operating profit , in millions?

Solution:
The text provides operating profits for 2008 (17 million) and 2009 (433 million).
Subtracting the 2008 amount from the 2009 amount: 433 - 17 = 416. The final answer is
\(\boxed{416}\).
------
Problem:
[[PROBLEM]]
```

**Figure 6**: CoT prompt used for FinQA (four in-domain demonstrations).

972
973
974
975
976
977
978
979
980
981
982
983
984
985
986
987
988
989
990
991
992
993
994
995
996
997
998
999
1000
1001
1002
1003
1004
1005
1006
1007
1008
1009
1010
1011
1012
1013
1014
1015
1016
1017
1018
1019
1020
1021
1022
1023
1024
1025

## PoT Prompt - FinQA

```
Please generate Python code to solve the given mathematical problem. The code should
first  define the input parameter values as stated in the problem, then provide the solution
code, and then store the final answer in a variable named 'ans'. Be mindful of units handling
in your solution.
------
Question: for uncoated freesheet paper and market pulp announced
at the end of 2009 become effective . input costs are expected to be higher due to wood
supply constraints at the kwidzyn mill and annual tariff increases on energy in russia .
planned main- tenance outage costs are expected to be about flat , while operating costs should
be favorable . asian printing papers net sales were approx- imately $ 50 million in 2009
compared with approx- imately $ 20 million in both 2008 and 2007 . operating earnings increased
slightly in 2009 compared with 2008 , but were less than $ 1 million in all periods . u.s .
market pulp net sales in 2009 totaled $ 575 million compared with $ 750 million in 2008 and $
655 million in 2007 . operating earnings in 2009 were $ 140 million ( a loss of $ 71 million
excluding alter- native fuel mixture credits and plant closure costs ) compared with a loss of $
156 million ( a loss of $ 33 million excluding costs associated with the perma- nent
shutdown of the bastrop mill ) in 2008 and earn- ings of $ 78 million in 2007 . sales volumes in
2009 decreased from 2008 levels due to weaker global demand . average sales price
realizations were significantly lower as the decline in demand resulted in significant price declines
for market pulp and smaller declines in fluff pulp . input costs for wood , energy and
chemicals decreased , and freight costs were significantly lower . mill operating costs were
favorable across all mills , and planned maintenance downtime costs were lower . lack-of-order
downtime in 2009 increased to approx- imately 540000 tons , including 480000 tons related to the
permanent shutdown of our bastrop mill in the fourth quarter of 2008 , compared with 135000 tons
in 2008 . in the first quarter of 2010 , sales volumes are expected to increase slightly ,
reflecting improving customer demand for fluff pulp , offset by slightly seasonally weaker demand
for softwood and hard- wood pulp in china . average sales price realizations are expected to
improve , reflecting the realization of previously announced sales price increases for fluff
pulp , hardwood pulp and softwood pulp . input costs are expected to increase for wood ,
energy and chemicals , and freight costs may also increase . planned maintenance downtime costs
will be higher , but operating costs should be about flat . consumer packaging demand and
pricing for consumer packaging prod- ucts correlate closely with consumer spending and general
economic activity . in addition to prices and volumes , major factors affecting the profitability
of consumer packaging are raw material and energy costs , freight costs , manufacturing
efficiency and product mix . consumer packaging net sales in 2009 decreased 4% ( 4 % ) compared
with 2008 and increased 1% ( 1 % ) compared with 2007 . operating profits increased
significantly compared with both 2008 and 2007 . excluding alternative fuel mixture credits and
facility closure costs , 2009 operating profits were sig- nificantly higher than 2008 and 57% (
57 % ) higher than 2007 . benefits from higher average sales price realizations ( $ 114
million ) , lower raw material and energy costs ( $ 114 million ) , lower freight costs ( $ 21
million ) , lower costs associated with the reorganiza- tion of the shorewood business ( $ 23
million ) , favor- able foreign exchange effects ( $ 14 million ) and other items ( $ 12 million
) were partially offset by lower sales volumes and increased lack-of-order downtime ( $
145 million ) and costs associated with the perma- nent shutdown of the franklin mill ( $ 67
million ) . additionally , operating profits in 2009 included $ 330 million of alternative fuel
mixture credits . consumer packaging in millions 2009 2008 2007 .
--------------------------------------------------------------
\n| in millions     | 2009   | 2008   | 2007   |
\n|-----------------|--------|--------|--------|
\n| sales           | $ 3060 | $ 3195 | $ 3015 |
\n| operating profit | 433   | 17     | 112    |
\n--------------------------------------------------------------
sales: 2009: $3060, 2008: $ 3195, 2007: $ 3015. operating profit: 2009: 433, 2008: 17, 2007:
112. north american consumer packaging net sales were $ 2.2 billion compared with $ 2.5 billion in 2008
and $ 2.4 billion in 2007 . operating earnings in 2009 were $ 343 million ( $ 87 million
excluding alter- native fuel mixture credits and facility closure costs ) compared with $ 8
million ( $ 38 million excluding facility closure costs ) in 2008 and $ 70 million in 2007 .
coated paperboard sales volumes were lower in 2009 compared with 2008 reflecting weaker market
conditions . average sales price realizations were significantly higher , reflecting the full-year
realization of price increases implemented in the second half of 2008 . raw material costs for wood
, energy and chemicals were significantly lower in 2009 , while freight costs were also
favorable . operating costs , however , were unfavorable and planned main- tenance downtime costs
were higher . lack-of-order downtime increased to 300000 tons in 2009 from 15000 tons in 2008
due to weak demand . operating results in 2009 include income of $ 330 million for alternative
fuel mixture credits and $ 67 million of expenses for shutdown costs for the  franklin mill .
foodservice sales volumes were lower in 2009 than in 2008 due to generally weak world-wide
economic conditions . average sales price realizations were . what is the average value for sales?
```

```
PythonCode:
# input parameters
sales_2009 = 3060
sales_2008 = 3195
sales_2007 = 3015

# solution code
ans = (sales_2009 + sales_2008 + sales_2007) / 3
------
Question:
interest rate cash flow hedges 2013 we report changes in the fair value of cash flow hedges in
accumulated other comprehensive loss until the hedged item affects earnings . at both december 31 ,
2008 and 2007 , we had reductions of $ 4 million recorded as an accumulated other
comprehensive loss that is being amortized on a straight-line basis through september 30 , 2014 . as
of december 31 , 2008 and 2007 , we had no interest rate cash flow hedges outstanding .
earnings impact 2013 our use of derivative financial instruments had the following impact on
pre-tax income for the years ended december 31 : millions of dollars 2008 2007 2006 .
-------------------------------------------------------------------------------
\n| millions of dollars                                          |2008 | 2007  | 2006    |
\n|------------------------------------------------------------|----|--------|---------|
\n| ( increase ) /decrease in interest expense from interest rate hedging |$ 1  | $ -8 (8) | $ -8 ( 8 ) |
\n| ( increase ) /decrease in fuel expense from fuel derivatives      | 1  |-1 ( 1 ) | 3       |
\n| increase/ ( decrease ) in pre-tax income                         | $ 2| $ -9 (9) | $ -5 (5)|
\n-------------------------------------------------------------------------------
( increase ) /decrease in interest expense from interest rate hedging: 2008: $ 1, 2007:
$ -8 ( 8 ), 2006: $ -8 ( 8 ). ( increase ) /decrease in fuel expense from fuel
derivatives: 2008: 1, 2007: -1 ( 1 ), 2006: 3. increase/ ( decrease ) in pre-tax income: 2008: $ 2,
2007: $ -9 ( 9 ), 2006: $ -5 ( 5 ). fair value of debt instruments 2013 the fair value of our
short- and long-term debt was estimated using quoted market prices , where available , or
current borrowing rates . at december 31 , 2008 , the fair value of total debt is approximately
$ 247 million less than the carrying value . at december 31 , 2007 , the fair value of
total debt exceeded the carrying value by approximately $ 96 million . at december 31 , 2008
and 2007 , approximately $ 320 million and $ 181 million , respectively , of fixed-rate debt
securities contained call provisions that allowed us to retire the debt instruments prior to final
maturity , with the payment of fixed call premiums , or in certain cases , at par . sale of
receivables 2013 the railroad transfers most of its accounts receivable to union pacific receivables
, inc . ( upri ) , a bankruptcy-remote subsidiary , as part of a sale of receivables
facility . upri sells , without recourse on a 364-day revolving basis , an undivided interest in
such accounts receivable to investors . the total capacity to sell undivided interests to
investors under the facility was $ 700 million and $ 600 million at december 31 , 2008 and 2007 ,
respectively . the value of the outstanding undivided interest held by investors under the facility
was $ 584 million and $ 600 million at december 31 , 2008 and 2007 , respectively . upri
reduced the outstanding undivided interest held by investors due to a decrease in available
receivables at december 31 , 2008 . the value of the outstanding undivided interest held by
investors is not included in our consolidated financial statements . the value of the undivided
interest held by investors was supported by $ 1015 million and $ 1071 million of accounts
receivable held by upri at december 31 , 2008 and 2007 , respectively . at december 31 , 2008 and
2007 , the value of the interest retained by upri was $ 431 million and $ 471 million ,
respectively . this retained interest is included in accounts receivable in our consolidated
financial statements . the interest sold to investors is sold at carrying value , which
approximates fair value , and there is no gain or loss recognized from the transaction . the value of
the outstanding undivided interest held by investors could fluctuate based upon the
availability of eligible receivables and is directly affected by changing business volumes and credit
risks , including default and dilution . if default or dilution percentages were to increase
one percentage point , the amount of eligible receivables would decrease by $ 6 million .
should our credit rating fall below investment grade , the value of the outstanding undivided
interest held by investors would be reduced , and , in certain cases , the investors would have
the right to discontinue the facility . the railroad services the sold receivables ; however
, the railroad does not recognize any servicing asset or liability as the servicing fees
adequately compensate us for these responsibilities . the railroad collected approximately $ 17.8
billion and $ 16.1 billion during the years ended december 31 , 2008 and 2007 , respectively .
upri used certain of these proceeds to purchase new receivables under the facility. . what
was the difference in billions of sold receivables from 2007 to 2008?

PythonCode:
# input parameters
receivables_2007 = 16.1  # billions of dollars
receivables_2008 = 17.8  # billions of dollars

# solution code
ans = receivables_2008 - receivables_2007
```

```
------
Question:
baker hughes , age company notes to consolidated and combined financial statements bhge 2017 form 10-k
| 83 issuance pursuant to awards granted under the lti plan over its term which expires on the date of
the annual meeting of the company in 2027 . a total of 53.7 million shares of class a common
stock are available for issuance as of december 31 , 2017 . as a result of the acquisition of
baker hughes , on july 3 , 2017 , each outstanding baker hughes stock option was converted
into an option to purchase a share of class a common stock in the company . consequently , we
issued 6.8 million stock options which are fully vested . each converted option is subject to
the same terms and conditions as applied to the original option , and the per share exercise
price of each converted option was reduced by $ 17.50 to reflect the per share amount of the
special dividend pursuant to the agreement associated with the transactions . additionally , as
a result of the acquisition of baker hughes , there were 1.7 million baker hughes
restricted stock units ( rsus ) that were converted to bhge rsus at a fair value of $ 40.18 .
stock-based compensation cost is measured at the date of grant based on the calculated fair value of
the award and is generally recognized on a straight-line basis over the vesting period of
the equity grant . the compensation cost is determined based on awards ultimately expected
to vest ; therefore , we have reduced the cost for estimated forfeitures based on
historical forfeiture rates . forfeitures are estimated at the time of grant and revised , if
necessary , in subsequent periods to reflect actual forfeitures . there were no stock-based
compensation costs capitalized as the amounts were not material . during the year ended december 31 ,
2017 , we issued 2.1 million rsus and 1.6 million stock options under the lti plan . these
rsus and stock options generally vest in equal amounts over a three-year vesting period
provided that the employee has remained continuously employed by the company through such vesting
date . stock based compensation expense was $ 37 million in 2017 . included in this amount is
$ 15 million of expense which relates to the acceleration of equity awards upon
termination of employment of baker hughes employees with change in control agreements , and are
included as part of \"merger and related costs\" in the consolidated and combined statements of
income ( loss ) . as bhge llc is a pass through entity , any tax benefit would be recognized by
its partners . due to its cumulative losses , bhge is unable to recognize a tax benefit on
its share of stock related expenses . stock options the fair value of each stock option
granted is estimated using the black-scholes option pricing model . the following table presents
the weighted average assumptions used in the option pricing model for options granted under
the lti plan . the expected life of the options represents the period of time the options
are expected to be outstanding . the expected life is based on a simple average of the
vesting term and original contractual term of the awards . the expected volatility is based on
the historical volatility of our five main competitors over a six year period . the
risk-free interest rate is based on the observed u.s . treasury yield curve in effect at the time
the options were granted . the dividend yield is based on a five year history of dividend
payouts in baker hughes. .
-----------------------------------------------------------------------
\n|                                                     | 2017         |
\n|-----------------------------------------------------|--------------|
\n| expected life ( years )                             | 6            |
\n| risk-free interest rate                             | 2.1% ( 2.1 % ) |
\n| volatility                                          | 36.4% ( 36.4 % ) |
\n| dividend yield                                      | 1.2% ( 1.2 % ) |
\n| weighted average fair value per share at grant date | $12.32       |
 \n-----------------------------------------------------------------------
expected life ( years ): 2017: 6. risk-free interest rate: 2017: 2.1% ( 2.1 % ).
volatility: 2017: 36.4% ( 36.4 % ). dividend yield: 2017: 1.2% ( 1.2 % ). weighted average fair
value per share at grant date: 2017: $ 12.32. what is the total value of rsus converted to
bhge rsus , in millions?

PythonCode:
# input parameters
num_rsus = 1.7      # in millions of units
fair_value = 40.18  # in dollars per share

# solution code
ans = num_rsus * fair_value
------
Question:
for uncoated freesheet paper and market pulp announced at the end of 2009 become
effective .  input costs are expected to be higher due to wood supply constraints at the kwidzyn
mill and annual tariff increases on energy in russia .planned main- tenance outage costs are
expected to be about flat , while operating costs should be favorable . asian printing papers net
sales were approx- imately $ 50 million in 2009 compared with approx- imately $ 20 million in
both 2008 and 2007 . operating earnings increased slightly in 2009 compared with 2008 , but
were less than $ 1 million in all periods . u.s . market pulp net sales in 2009 totaled $ 575
```

```
million compared with $ 750 million in 2008 and $ 655 million in 2007 . operating earnings in
2009 were $ 140 million ( a loss of $ 71 million excluding alter- native fuel mixture credits
and plant closure costs ) compared with a loss of $ 156 million ( a loss of $ 33 million
excluding costs associated with the perma- nent shutdown of the bastrop mill ) in 2008 and earn-
ings of $ 78 million in 2007 . sales volumes in 2009 decreased from 2008 levels due to weaker
global demand . average sales price realizations were significantly lower as the decline in
demand resulted in significant price declines for market pulp and smaller declines in fluff
pulp . input costs for wood , energy and chemicals decreased , and freight costs were
significantly lower . mill operating costs were favorable across all mills , and planned maintenance
downtime costs were lower . lack-of-order downtime in 2009 increased to approx- imately 540000
tons , including 480000 tons related to the permanent shutdown of our bastrop mill in the
fourth quarter of 2008 , compared with 135000 tons in 2008 . in the first quarter of 2010 ,
sales volumes are expected to increase slightly , reflecting improving customer demand for
fluff pulp , offset by slightly seasonally weaker demand for softwood and hard- wood pulp in
china . average sales price realizations are expected to improve , reflecting the realization
of previously announced sales price increases for fluff pulp , hardwood pulp and softwood
pulp . input costs are expected to increase for wood , energy and chemicals , and freight
costs may also increase . planned maintenance downtime costs will be higher , but operating
costs should be about flat . consumer packaging demand and pricing for consumer packaging
prod- ucts correlate closely with consumer spending and general economic activity . in
addition to prices and volumes , major factors affecting the profitability of consumer packaging
are raw material and energy costs , freight costs , manufacturing efficiency and product mix
. consumer packaging net sales in 2009 decreased 4% ( 4 % ) compared with 2008 and
increased 1% ( 1 % ) compared with 2007 . operating profits increased significantly compared with
both 2008 and 2007 . excluding alternative fuel mixture credits and facility closure costs ,
2009 operating profits were sig- nificantly higher than 2008 and 57% ( 57 % ) higher than
2007 . benefits from higher average sales price realizations ( $ 114 million ) , lower raw
material and energy costs ( $ 114 million ) , lower freight costs ( $ 21 million ) , lower costs
associated with the reorganiza- tion of the shorewood business ( $ 23 million ) , favor- able
foreign exchange effects ( $ 14 million ) and other items ( $ 12 million ) were partially offset
by lower sales volumes and increased lack-of-order downtime ( $ 145 million ) and costs
associated with the perma- nent shutdown of the franklin mill ( $ 67 million ) . additionally ,
operating profits in 2009 included $ 330 million of alternative fuel mixture credits . consumer
packaging in millions 2009 2008 2007 .
----------------------------------------------------------------
\n| in millions      | 2009   | 2008   | 2007   |
\n|------------------|--------|--------|--------|
\n| sales            | $ 3060 | $ 3195 | $ 3015 |
\n| operating profit | 433    | 17     | 112    |
\n----------------------------------------------------------------
sales: 2009: $3060, 2008: $ 3195, 2007: $ 3015. operating profit: 2009: 433, 2008: 17, 2007: 112.
north american consumer packaging net sales were $ 2.2 billion compared with $ 2.5 billion in 2008 and
$ 2.4 billion in 2007 . operating earnings in 2009 were $ 343 million ( $ 87 million
excluding alter- native fuel mixture credits and facility closure costs ) compared with $ 8
million ( $ 38 million excluding facility closure costs ) in 2008 and $ 70 million in 2007 .
coated paperboard sales volumes were lower in 2009 compared with 2008 reflecting weaker market
conditions . average sales price realizations were significantly higher , reflecting the full-year
realization of price increases implemented in the second half of 2008 . raw material costs for wood
, energy and chemicals were significantly lower in 2009 , while freight costs were also
favorable . operating costs , however , were unfavorable and planned main- tenance downtime costs
were higher . lack-of-order downtime increased to 300000 tons in 2009 from 15000 tons in 2008
due to weak demand . operating results in 2009 include income of $ 330 million for
alternative fuel mixture credits and $ 67 million of expenses for shutdown costs for the franklin
mill . foodservice sales volumes were lower in 2009 than in 2008 due to generally weak
world-wide economic conditions . average sales price realizations were . considering the years 2008
and 2009 , what is the variation observed in the operating profit , in millions?

PythonCode:
# input parameters
operating_profit_2009 = 433  # in millions of dollars
operating_profit_2008 = 17   # in millions of dollars

# solution code
ans = operating_profit_2009 - operating_profit_2008
------
Question:
[[QUESTION]]
PythonCode:
```

**Figure 7**: PoT prompt used for FinQA (four in-domain demonstrations).

```
CoT Prompt - TabMWP

Please solve the given mathematical problem, doing step by step reasoning
to arrive at the final answer. Please mark the final answer in a "\\boxed{}"
annotation as shown in the example below.
------
Problem:
A bus driver paid attention to how many passengers her bus had each day.
On which day did the bus have the fewest passengers?

People on the bus:
Day | Number of people
Monday | 39
Tuesday | 38
Wednesday | 32
Thursday | 36

Solution:
Find the least number in the table. Remember to compare the numbers
tarting with the highest place value. The least number is 32.

Now find the corresponding day. Wednesday corresponds to 32.
The final answer is \boxed{Wednesday}.

------

Problem:
Jayla has $95.35. How much money will Jayla have left if she buys a CD player
and a DVD?

None:
CD | $13.37
CD player | $16.61
DVD | $13.28
alarm clock | $13.72
microwave | $53.38

Solution:
Find the total cost of a CD player and a DVD.

$16.61 + $13.28 = $29.89

Now subtract the total cost from the tarting amount.

$95.35 - $29.89 = $65.46

Jayla will have $65.46 left. The final answer is \boxed{65.46 $}.
------
Problem:
[[PROBLEM]]
```

**Figure 8**: CoT prompt used for TabMWP (two in-domain demonstrations).

```
PoT Prompt - TabMWP

Please generate Python code to solve the given mathematical problem. The
code should store the final answer in a variable named 'ans' as shown in the
example below.
------
Question:
A bus driver paid attention to how many passengers her bus had each day. On
which  day did the bus have the fewest passengers?

People on the bus:
Day | Number of people
Monday | 39
Tuesday | 38
Wednesday | 32
Thursday | 36
PythonCode:
bus_data = {
    "Monday": 39,
    "Tuesday": 38,
    "Wednesday": 32,
    "Thursday": 36
}

min_passengers = float('inf')
ans = None

for day, passengers in bus_data.items():
    if passengers < min_passengers:
        min_passengers = passengers
        ans = day
------
Question:
Jayla has $95.35. How much money will Jayla have left if she buys a CD player
and a DVD?

None:
CD | $13.37
CD player | $16.61
DVD | $13.28
alarm clock | $13.72
microwave | $53.38
PythonCode:
# Input parameters
cd_player_cost = 16.61
dvd_cost      = 13.28
initial_amount = 95.35

# Solution
ans = initial_amount - (cd_player_cost + dvd_cost)
------
Question:
[[QUESTION]]
PythonCode:
```

**Figure 9**: PoT prompt used for TabMWP (two in-domain demonstrations).

## CoT-to-PoT Generation Prompt for Boostrapping PoT Rationales

Please generate Python code to solve the given mathematical problem. The code should first define the input parameter values as stated in the problem, then provide the solution code, and then store the final answer in a variable named 'ans'. The first 8 question and python code pairs are given as an example, solve the last question.

------
Question:
There are 15 trees in the grove. Grove workers will plant trees in the grove today. After they are done, there will be 21 trees. How many trees did the grove workers plant today?
Solution:
There are 15 trees originally. Then there were 21 trees after some more were planted. So there must have been 21 - 15 = 6. The final answer is \\boxed{6}
PythonCode:
```
# input parameters
initial_trees = 15
final_trees = 21

# solution code
ans = final_trees - initial_trees
```
------
Question:
If there are 3 cars in the parking lot and 2 more cars arrive, how many cars are in the parking lot?
Solution:
There are originally 3 cars. 2 more cars arrive. 3 + 2 = 5. The final answer is \\boxed{5}
PythonCode:
```
# input parameters
initial_cars = 3
arrived_cars = 2

# solution code
ans = initial_cars + arrived_cars
```
------
Question:
Leah had 32 chocolates and her sister had 42. If they ate 35, how many pieces do they have left in total?
Solution:
Originally, Leah had 32 chocolates. Her sister had 42. So in total they had 32 + 42 = 74. After eating 35, they had 74 - 35 = 39. The final answer is \\boxed{39}
PythonCode:
```
# input parameters
leah_chocolates = 32
sister_chocolates = 42
eaten_chocolates = 35

# solution code
total_chocolates = leah_chocolates + sister_chocolates
ans = total_chocolates - eaten_chocolates
```
------
Question:
Jason had 20 lollipops. He gave Denny some lollipops. Now Jason has 12 lollipops. How many lollipops did Jason give to Denny?
Solution:
Jason started with 20 lollipops. Then he had 12 after giving some to Denny. So he gave Denny 20 - 12 = 8. The final answer is \\boxed{8}
PythonCode:
```
# input parameters
initial_lollipops = 20
remaining_lollipops = 12

# solution code
ans = initial_lollipops - remaining_lollipops
```
------
Question:
Shawn has five toys. For Christmas, he got two toys each from his mom and dad. How many toys does he have now?
Solution:
Shawn started with 5 toys. If he got 2 toys each from his mom and dad, then that is 4 more toys. 5 + 4 = 9. The final answer is \\boxed{9}

```
PythonCode:
# input parameters
initial_toys = 5
toys_from_mom = 2
toys_from_dad = 2

# solution code
ans = initial_toys + toys_from_mom + toys_from_dad
------
Question:
There were nine computers in the server room. Five more computers were installed each day, from Monday
to thursday. How many computers are now in the server room?
Solution:
There were originally 9 computers. For each of 4 days, 5 more computers were added. So 5 * 4 = 20
computers were added. 9 + 20 is 29. The final answer is \\boxed{29}
PythonCode:
# input parameters
initial_computers = 9
additional_computers_per_day = 5
days = 4

# solution code
total_additional_computers = additional_computers_per_day * days
ans = initial_computers + total_additional_computers
------
Question:
Michael had 58 golf balls. On tuesday, he lost 23 golf balls. On wednesday, he lost 2 more. How many
golf balls did he have at the end of wednesday?
Solution:
Michael started with 58 golf balls. After losing 23 on tuesday, he had 58 - 23 = 35. After losing 2
more, he had 35 - 2 = 33 golf balls. The final answer is \\boxed{33}
PythonCode:
# input parameters
initial_golf_balls = 58
lost_golf_balls_tuesday = 23
lost_golf_balls_wednesday = 2

# solution code
remaining_golf_balls = initial_golf_balls - lost_golf_balls_tuesday
ans = remaining_golf_balls - lost_golf_balls_wednesday
------
Question:
Olivia has $23. She bought five bagels for $3 each. How much money does she have left?
Solution:
Olivia had 23 dollars. 5 bagels for 3 dollars each will be 5 x 3 = 15 dollars. So she has 23 - 15
dollars left. 23 - 15 is 8. The final answer is \\boxed{8}
PythonCode:
# input parameters
initial_money = 23
bagel_cost = 3
num_bagels = 5

# solution code
total_cost = bagel_cost * num_bagels
ans = initial_money - total_cost
------
Question:
[[QUESTION]]
Solution:
[[SOLUTION]]
PythonCode:
```

**Figure 10**: Prompt used for generation of PoT rationales from CoT in Case Study.

## A.2 FULL TABLE ENSEMBLE RESULTS

### A.2.1 FULL-SAMPLING RESULTS

This section reports the complete numbers for all full-sampling variants for each dataset and model separately.

Tables 6 - 9 give per-dataset accuracies for GPT-3.5, GPT-4o, MISTRAL-LARGE and Qwen3-Coder. The final row reports the average across the five benchmarks.[1]

| Dataset | SC$_{CoT}$ | CP$_{Maj}$ | CP$_{Max}$ | CP$_{Agr}$ | SC$_{PoT}$ |
|---|---|---|---|---|---|
| GSM8K | 89.4 | 91.4 | 90.6 | 90.8 | 82.0 |
| MATH | 52.6 | 56.4 | 56.2 | 55.4 | 43.6 |
| SVAMP | 91.4 | 93.1 | 92.4 | 93.1 | 89.0 |
| FINQA | 57.0 | 60.0 | 60.2 | 60.0 | 58.0 |
| TABMWP | 78.8 | 77.6 | 77.2 | 77.6 | 72.8 |
| **Average** | 73.8 | 75.7 | 75.3 | 75.4 | 69.1 |

Table 6: Full-sampling accuracy (%) on **GPT-3.5**.

| Dataset | SC$_{CoT}$ | CP$_{Maj}$ | CP$_{Max}$ | CP$_{Agr}$ | SC$_{PoT}$ |
|---|---|---|---|---|---|
| GSM8K | 97.8 | 97.6 | 97.8 | 97.6 | 97.0 |
| MATH | 77.8 | 79.6 | 79.4 | 79.2 | 69.0 |
| SVAMP | 95.5 | 96.6 | 96.2 | 96.6 | 96.6 |
| FINQA | 63.4 | 63.0 | 63.0 | 62.8 | 64.4 |
| TABMWP | 82.8 | 88.8 | 88.8 | 89.8 | 88.4 |
| **Average** | 83.5 | 85.1 | 85.0 | 85.2 | 83.1 |

Table 7: Full-sampling accuracy (%) on **GPT-4o**.

| Dataset | SC$_{CoT}$ | CP$_{Maj}$ | CP$_{Max}$ | CP$_{Agr}$ | SC$_{PoT}$ |
|---|---|---|---|---|---|
| GSM8K | 97.2 | 97.2 | 97.2 | 97.4 | 96.4 |
| MATH | 73.2 | 77.2 | 77.2 | 77.0 | 66.4 |
| SVAMP | 94.5 | 95.5 | 95.5 | 95.2 | 94.8 |
| FINQA | 63.4 | 64.6 | 64.8 | 64.6 | 64.8 |
| TABMWP | 77.6 | 83.0 | 83.2 | 83.6 | 89.4 |
| **Average** | 81.2 | 83.5 | 83.6 | 83.6 | 82.4 |

Table 8: Full-sampling accuracy (%) on **Mistral-large**.

| Dataset | SC$_{CoT}$ | CP$_{Maj}$ | CP$_{Max}$ | CP$_{Agr}$ | SC$_{PoT}$ |
|---|---|---|---|---|---|
| GSM8K | 96.0 | 96.2 | 96.2 | 96.2 | 96.0 |
| MATH | 86.2 | 85.8 | 86.2 | 86.2 | 77.8 |
| SVAMP | 96.9 | 96.9 | 96.9 | 96.9 | 96.6 |
| FINQA | 58.8 | 62.8 | 62.4 | 62.8 | 64.4 |
| TABMWP | 79.4 | 87.8 | 87.8 | 88.2 | 90.8 |
| **Average** | 83.5 | 85.9 | 85.9 | 86.1 | 85.1 |

Table 9: Full-sampling accuracy (%) on **Qwen3-Coder**.

### A.2.2 EARLY-STOPPING RESULTS

We provide full results for early stopping for each dataset and model. Tables 10 - 12 list accuracies and average sample budgets for all models.

---

[1]GSM8K, MATH, SVAMP, FinQA, and TabMWP.

| Dataset | Accuracy (%) | | | | | | | Avg. # Samples | | | | | | |
|---|---|---|---|---|---|---|---|---|---|---|---|---|---|---|
| | ASC | A$_{ASC-CP}$ | CP$_{AA}$ | CP$_{FA}$ | CP$_{FF}$ | CP$_{DAA}$ | CP$_{DFA}$ | ASC | A$_{ASC-CP}$ | CP$_{AA}$ | CP$_{FA}$ | CP$_{FF}$ | CP$_{DAA}$ | CP$_{DFA}$ |
| GSM8K | 89.4 | 91.4 | 89.8 | 90.6 | 91.4 | 90.6 | 90.4 | 8.5 | 10.8 | 4.2 | 5.3 | 8.4 | 4.9 | 6.9 |
| MATH | 52.6 | 56.2 | 55.2 | 54.6 | 56.6 | 55.6 | 55.8 | 22.9 | 24.3 | 13.4 | 17.5 | 23.2 | 13.7 | 18.6 |
| SVAMP | 91.4 | 93.1 | 91.0 | 92.4 | 92.8 | 91.0 | 92.4 | 8.3 | 8.0 | 3.6 | 4.0 | 6.0 | 3.6 | 4.9 |
| FINQA | 56.8 | 60.0 | 59.2 | 59.4 | 60.0 | 59.4 | 59.8 | 12.2 | 14.3 | 6.3 | 7.3 | 12.4 | 6.3 | 8.6 |
| TABMWP | 79.0 | 77.4 | 78.4 | 78.6 | 77.6 | 78.4 | 78.0 | 9.9 | 10.2 | 4.2 | 4.8 | 8.0 | 4.3 | 5.5 |
| **Average** | 73.8 | 75.6 | 74.7 | 75.1 | 75.7 | 75.0 | 75.3 | 12.4 | 13.5 | 6.3 | 7.8 | 11.6 | 6.5 | 8.9 |

Table 10: Accuracy and average sample budget for **GPT-3.5**.

| Dataset | Accuracy (%) | | | | | | | Avg. # Samples | | | | | | |
|---|---|---|---|---|---|---|---|---|---|---|---|---|---|---|
| | ASC | A$_{ASC-CP}$ | CP$_{AA}$ | CP$_{FA}$ | CP$_{FF}$ | CP$_{DAA}$ | CP$_{DFA}$ | ASC | A$_{ASC-CP}$ | CP$_{AA}$ | CP$_{FA}$ | CP$_{FF}$ | CP$_{DAA}$ | CP$_{DFA}$ |
| GSM8K | 97.8 | 97.6 | 96.8 | 96.6 | 97.6 | 96.8 | 96.6 | 4.6 | 4.9 | 2.3 | 2.5 | 2.9 | 2.3 | 2.5 |
| MATH | 77.6 | 79.6 | 78.8 | 78.2 | 79.6 | 78.8 | 78.6 | 11.3 | 13.9 | 7.5 | 9.0 | 12.0 | 7.5 | 9.4 |
| SVAMP | 95.5 | 96.6 | 96.2 | 96.2 | 96.2 | 96.2 | 96.2 | 5.0 | 4.9 | 2.3 | 2.3 | 2.6 | 2.3 | 2.3 |
| FINQA | 63.4 | 63.0 | 62.8 | 63.2 | 63.0 | 62.8 | 62.6 | 6.8 | 7.2 | 3.3 | 3.4 | 5.1 | 3.3 | 4.5 |
| TABMWP | 82.8 | 88.8 | 89.4 | 89.2 | 88.8 | 89.4 | 89.2 | 6.1 | 9.1 | 6.3 | 6.4 | 7.5 | 6.3 | 6.4 |
| **Average** | 83.4 | 85.1 | 84.8 | 84.7 | 85.0 | 84.8 | 84.6 | 6.8 | 8.0 | 4.3 | 4.7 | 6.0 | 4.3 | 5.0 |

Table 11: Accuracy and average sample budget for **GPT-4o**.

| Dataset | Accuracy (%) | | | | | | | Avg. # Samples | | | | | | |
|---|---|---|---|---|---|---|---|---|---|---|---|---|---|---|
| | ASC | A$_{ASC-CP}$ | CP$_{AA}$ | CP$_{FA}$ | CP$_{FF}$ | CP$_{DAA}$ | CP$_{DFA}$ | ASC | A$_{ASC-CP}$ | CP$_{AA}$ | CP$_{FA}$ | CP$_{FF}$ | CP$_{DAA}$ | CP$_{DFA}$ |
| GSM8K | 97.4 | 97.2 | 97.8 | 97.8 | 97.6 | 97.8 | 97.8 | 4.5 | 4.7 | 2.2 | 2.2 | 2.5 | 2.2 | 2.2 |
| MATH | 73.0 | 77.2 | 75.2 | 76.0 | 77.0 | 76.2 | 76.8 | 12.5 | 14.8 | 7.8 | 9.5 | 13.0 | 8.1 | 11.0 |
| SVAMP | 94.5 | 95.5 | 94.5 | 94.5 | 95.5 | 94.5 | 95.5 | 5.1 | 5.3 | 2.3 | 2.5 | 3.0 | 2.3 | 2.6 |
| FINQA | 63.4 | 64.4 | 64.6 | 64.4 | 63.8 | 64.4 | 64.2 | 7.5 | 8.1 | 3.3 | 3.7 | 5.4 | 3.3 | 4.3 |
| TABMWP | 77.6 | 83.0 | 83.0 | 83.0 | 82.6 | 83.0 | 83.0 | 5.0 | 12.0 | 9.5 | 9.5 | 10.3 | 9.5 | 9.5 |
| **Average** | 81.2 | 83.5 | 83.0 | 83.1 | 83.3 | 83.2 | 83.5 | 6.9 | 9.0 | 5.0 | 5.5 | 6.8 | 5.1 | 5.9 |

Table 12: Accuracy and average sample budget for **Mistral-large**.

| Dataset | Accuracy (%) | | | | | | | Avg. # Samples | | | | | | |
|---|---|---|---|---|---|---|---|---|---|---|---|---|---|---|
| | ASC | A$_{ASC-CP}$ | CP$_{AA}$ | CP$_{FA}$ | CP$_{FF}$ | CP$_{DAA}$ | CP$_{DFA}$ | ASC | A$_{ASC-CP}$ | CP$_{AA}$ | CP$_{FA}$ | CP$_{FF}$ | CP$_{DAA}$ | CP$_{DFA}$ |
| FINQA | 58.8 | 63.0 | 62.2 | 62.4 | 62.4 | 62.2 | 62.6 | 7.6 | 10.2 | 4.8 | 4.9 | 7.6 | 5.0 | 6.9 |
| TABMWP | 79.6 | 87.8 | 88.0 | 87.8 | 87.8 | 88.0 | 87.8 | 6.1 | 10.0 | 6.2 | 6.5 | 8.4 | 6.2 | 6.5 |
| SVAMP | 96.9 | 96.9 | 96.9 | 96.9 | 96.9 | 96.9 | 96.9 | 4.3 | 4.4 | 2.3 | 2.3 | 2.4 | 2.3 | 2.3 |
| GSM8K | 95.8 | 96.2 | 96.4 | 96.4 | 96.4 | 96.4 | 96.2 | 4.8 | 5.0 | 2.3 | 2.3 | 2.8 | 2.3 | 2.4 |
| MATH | 86.2 | 85.8 | 86.6 | 86.8 | 85.8 | 86.6 | 86.0 | 9.3 | 11.0 | 6.9 | 7.6 | 9.2 | 6.9 | 8.4 |
| **Average** | 83.5 | 85.9 | 86.0 | 86.1 | 85.9 | 86.0 | 85.9 | 6.4 | 8.1 | 4.5 | 4.7 | 6.1 | 4.5 | 5.3 |

Table 13: Accuracy and average sample budget for **Qwen3-Coder**.

