# OpenReview forum: "When Two is Enough: CoT–PoT Ensembling for Efficient Self-Consistency in LLM Reasoning"
_ICLR.cc/2026/Conference — ICLR 2026 Conference Withdrawn Submission_

### Official Review · Reviewer_r1PH · 2025-10-27

**Soundness:** 2
**Presentation:** 3
**Contribution:** 2
**Rating:** 4
**Confidence:** 3

**Summary:**

- Paper proposes **cross-modal self-consistency**: sample CoT and PoT, aggregate under full budget (CPMaj/CPMax/CPAgr) or **early-stop** using a **Bayesian agreement** model; also gives simple $a_2=1$ heuristics (CPAA/CPFA/CPFF).
- Empirically improves over CoT-only/PoT-only self-consistency on math/tabular QA while **reducing samples** (often to two: one CoT + one PoT).

**Strengths:**

- **Clear practical goal:** leverage complementary error profiles of CoT vs PoT.
- **Simple deployable heuristics:** $a_2{=}1$ rules are easy to implement and often efficient.
- **Reasonable empirical coverage:** multiple datasets/models; ablations touch intra- vs cross-modal agreement.

**Weaknesses:**

- **Novelty:** Section 3.1 largely amounts to ensembling CoT and PoT with voting variants; the main gain appears to come from cross-modal diversity, which is conceptually incremental.
- **Narrow task scope:** all benchmarks reduce to numeric/program-executable answers; unclear applicability to open-ended reasoning.
- **Decorative theory:** Section 2.2.1’s Bayesian agreement model seems ornamental; best-performing procedures reduce to ad-hoc cross-modal agreement checks (CPAA/CPFA/CPFF with $a_2=1$).
- **Fairness:** Comparisons mix tool-using PoT vs tool-free CoT, no baseline with tool-augmented CoT under identical stopping.
- **Sensitivity & tuning:** Early-stopping hinges on thresholds / priors (e.g., $ρ$, $a_2$); robustness is not convincingly demonstrated.
- **Scope & metrics:** Only executable numeric tasks; no latency/token-cost reporting; limited statistical rigor (no CIs/significance).

**Questions:**

- **Necessity of Section 2.2.1:** What does the Bayesian agreement model add beyond the simple cross-modal agreement heuristics actually used (CPAA/CPFA/CPFF)? Can you show any setting where the Bayesian estimator changes decisions and improves over those heuristics?
- **Two-sample claim robustness:** For the “two is enough” claim, provide per-dataset/model histograms of stop counts and accuracy conditioned on early stop vs continued sampling. How stable is the ~2-sample regime under domain shift or weaker models?
- **Generality:** any results on non-executable, free-form reasoning (e.g., multi-hop QA, scientific explanations)?
- **Fair baselines:** calculator-augmented CoT and PoT-only early-stopping under identical thresholds?
- **Sensitivity:** curves for accuracy/efficiency vs $a_2$, $\rho$, temperature, and max budget; robustness when $a_2<1$?
- **Cost & latency:** token counts and wall-clock (including interpreter overhead) per dataset/model.
- **Failure modes:** when CoT and PoT agree yet are wrong, what patterns dominate, and can agreement be qualified (e.g., sanity checks)?

---

> ### Author Response · Authors · 2025-11-18
> **Response to review  r1PH [1/3]**
>
> We thank the reviewer for their very valuable feedback.
>
> **W1 (Novelty - Section 3.1 largely amounts to ensembling CoT and PoT with voting variants)**
>
> We agree that the full-sampling variants in Section 3.1 are conceptually and even empirically incremental in accuracy gains. However, this section is by no means the main contribution of our work and only small contribution (additional to our main contribution on drastic efficiency gains from cross-modality) and  is also important to include as a prerequisite for the following reasons: (1) we needed to establish that full sampling across the two modes does indeed provide higher accuracy rather than lower or similar (these results and voting techniques are not provided anywhere in the current literature to our knowledge)  (2) the full sampling methods of 3.1 establish the *base case* against which we compare our main results on the huge efficiency gains achieved with our early stopping methods (in section 3.2). We show the deltas of the accuracies against the best full sampling method for which we refer to the results in section 3.1.
>
> Thus, full-sampling CoT–PoT ensembling is not presented as a standalone innovation, but as a necessary base case that establishes why cross-modal signals are sufficiently strong to support early stopping with only two samples. We will clarify this motivation and positioning to avoid confusion, thanks.
>
> **W2 (Narrow task scope)**:
>
> We agree that the benefits of our approach are strongest for tasks that admit some form of programmatic or symbolic decomposition (which includes most mathematical reasoning tasks), but we do not believe it is a very narrow scope as it aligns with the large class of reasoning tasks targeted by numerous popular program-based and tool-augmented reasoning methods in the literature (e.g. Tora, ReTool, PAL etc.  which are also not targeted to or  evaluated in other domains outside of mathematical reasoning).  Our method is designed for and best suited to this broad category, and  the same limitation holds for any program-based LLM reasoning systems in the literature.  For tasks that are not naturally expressible as programs, the PoT channel will contribute less signal, but the framework still functions: PoT samples may still provide partial structure (e.g., arithmetic steps, helper variables) and non-programmatic reasoning may still be done in commented code, and disagreement prevents early stopping. We acknowledge that such non-programmatic tasks lie outside the main scope of our experiments and we have not evaluated on this, and we will clarify this scope and the intended applicability domain in the text.
>
> **W3 (Decorative theory - Bayesian agreement model seems ornamental)**
>
> Our Bayesian agreement has both theoretical and practical benefits. Firstly, it provides a theoretical foundation for early stopping based on cross-modal agreement, as we define the core notions of early stopping and *safety* probabilities, and illustrate how the  heuristic methods are special cases of this general  model when we choose extreme instantiations (without this it will be less clear why the heuristics methods work or what variations are possible, etc). Secondly, even in practical terms the data-driven methods are better than the heuristic ones in certain situations: for example, Table 3 shows that for the Mistral model overall, the most accurate method across all datasets is the data-driven CP_DFA method (which  in this case is more accurate than even the most conservate heuristic method and much more efficient than it). Hence if someone were using the Mistral model for deployment in practice then their best choice would be CP_DFA. Similarly, as we discuss in section 3.4, for the Llama model the best method is CP_DAA that is the most efficient method that beats the baseline accuracy. Hence in certain practical situations the data-driven instantiations that directly utilize our Bayesian model will be preferred. However, the advantage of the heuristic methods is that they perform well on average and offer a simple alternative that does not necessitate explicit pre-training.

---

> ### Author Response · Authors · 2025-11-18
> **Response to review r1PH [2/3]**
>
> **W4 (Fairness - no baseline with tool-augmented CoT)**
>
> We first note that PoT in our framework already plays a similar role to tool-augmented CoT— LLMs often  produce PoT programs that include CoT-style reasoning in natural-language comments alongside executable code, which effectively blends tool use with CoT-style reasoning. For instance, in some cases in advanced domains in the MATH dataset such as geometry or calculus where programmatic support is not available, we observe that PoTs contain most of the reasoning as code comments in natural language CoT to compute the answer and return it as a trivial program.  As for standard tool-augmented CoT methods (e.g., ReAct-style or calculator-augmented reasoning) typically require multiple interleaved LLM calls, multiple tool calls, or explicit orchestration, making each “sample” a multi-step trajectory rather than a single atomic draw. This makes their incorporation and analysis less clear in the  standard self-consistency setting with one shot prompting and comparing efficiency against such multi-step agents is also more difficult. But we agree that  explicitly comparing against more complex tool-augmented or agent-style frameworks is an interesting direction and will highlight it as future work.
>
>
> **W5 (Sensitivity & tuning)**
>
> Our early-stopping methods depend on only a few simple scalar parameters (e.g., the confidence threshold \rho and the priors c, a_1, a_2), and empirically we observe that the method is not highly sensitive to their exact values. Across four large models and five datasets, the data-independent extreme a_2 = 1 already yields strong performance, indicating broad robustness. When this assumption weakens—as in our Llama 3B case study—the data-driven variant automatically outperforms the fixed one, showing that the framework adapts when priors mismatch. Because these parameters are easy to estimate from a small calibration set (tens of examples), practical tuning is lightweight.
>
> **W6 (Scope & metrics)**
>
> See W2 above on our focus and scope on numeric (mathematical) reasoning tasks.  We agree that reporting latency and token costs would strengthen the empirical picture; although our method already provides large reductions in number of samples (which correlates strongly with cost), we will include concrete runtime/token measurements in the final version. Finally, we did not perform multiple repeat runs for   confidence intervals and significance testing  due to time and cost constraints, since self consistency experiments are highly expensive – we do note that prior SoTA works on self consistency efficiency also do not perform such repeat experiments for confidence reporting (agarwal et al 2023, Li et al., 2024).
>
> **Q1 (Necessity of Section 2.2.1)**
>
> Please see our response to [W3].
>
> **Q2 (Two-sample claim robustness)**:
>
> We already report the proportion of problems solved with two samples across all datasets and models (Fig. 2), showing that the 2-sample regime is consistent and high (≈76% on average, >90% in some settings). We agree that per-dataset/model histograms and conditional accuracies (early-stop vs continued sampling) would make this even more explicit, and we will add these plots and breakdowns in the revision. Under domain/model shift, our LLaMA-3B case study already represents a “weaker” setting, where the two-sample proportion and a_2 decrease, and performance is stabilized by the data-driven variants—indicating that the 2-sample regime remains useful but naturally degrades when cross-modal agreement is weaker. We will clarify this behavior and explicitly separate results conditioned on early vs late stopping.
>
> **Q3 (Generality)**
>
> Please see our response to [W2] above on our scope. Our focus is on mathematical reasoning tasks as is the case for many existing SoTA works that leverage programmatic tool use in LLM reasoning (Tora, Retool, PAL, etc). These approaches are not targeted to or  evaluated in free-form reasoning domains such as multi-hop QA, scientific explanations.
>
> **Q4 (Fair baselines)**
>
> Please see W4 above.
>
> **Q5 (Sensitivity)**
>
> We have included an analysis of efficiency vs max sampling budget (Figure 3) which shows the gains of our approach increase over baseline approaches as the budget increases. We can include similar curves for accuracy/efficiency vs , , temperature, and max budget.

---

> ### Author Response · Authors · 2025-11-18
> **Response to review r1PH [3/3]**
>
> **Q6 (Cost & latency)**
>
> We will report token counts (we expect similar patterns as the sampling gains themselves are significant and token counts are similar for cot and pot outputs). We will also report wall-clock though we did not notice significant interpreter overhead for program execution in such tasks.
>
> **Q7 (Failure modes)**
>
> We agree illustration of  incorrect agreement cases will be interesting and we will add some specific cases for illustration. Empirically, these events are rare as we have shown and when they do occur, they typically arise from shared misinterpretations of the problem statement (e.g., reading the wrong quantity, or adopting the same incorrect simplifying assumption or formula, etc).  Although our Bayesian treatment already treats agreement as a probabilistic signal rather than a guarantee, we acknowledge that additional lightweight checks (e.g., verifying program execution traces, numerical range checks, or problem-dependent sanity constraints) could further qualify agreement. We will add a brief discussion on these patterns and mention such sanity checks as potential natural extensions to the framework.

---

### Official Review · Reviewer_FMp9 · 2025-10-28

**Soundness:** 3
**Presentation:** 3
**Contribution:** 3
**Rating:** 4
**Confidence:** 4

**Summary:**

This paper studies how to improve self-consistency (SC) reasoning in large language models by combining two distinct reasoning modalities: Chain-of-Thought (CoT) and Program-of-Thought (PoT). The authors argue that CoT and PoT exhibit complementary error patterns—CoT being more flexible but error-prone in arithmetic, while PoT being computationally precise but symbolically brittle. They formalize the cross-modal agreement between these reasoning modalities through a Bayesian framework and propose both full-sampling and early-stopping strategies based on this formulation. Extensive experiments on multiple reasoning benchmarks and LLMs show that the approach achieves comparable or higher accuracy with far fewer samples, often requiring only two (one CoT and one PoT).

**Strengths:**

**Intuitive complementarity between reasoning modalities:**
Combining a natural-language stepwise reasoning modality (CoT) with a symbolic/programmatic modality (PoT) aligns with an intuitive notion of complementary error modes and increased diversity of reasoning traces. The framework captures this complementarity succinctly and leverages it for more efficient ensemble decisions.

**Potentially general framework for multimodal or heterogeneous reasoning:**
The paper presents a coherent probabilistic (Bayesian) framework that operationalizes how cross-modal agreement can be interpreted as a confidence signal. Although the experiments focus on CoT and PoT, the proposed Bayesian agreement mechanism could, in principle, be extended to any cross-modality reasoning setting, where agreement between different reasoning forms (e.g., textual, symbolic, or formal) can serve as a confidence signal for ensemble consistency.

**Empirical performance:**
 The method achieves remarkable efficiency improvements, solving the majority of tasks with only two samples while maintaining high accuracy.

**Weaknesses:**

**Assumption about program decomposability / limits of PoT usage.**
The approach leverages PoT outputs as one reasoning modality and implicitly assumes that tasks can be represented or approximated programmatically. While the Case Study discusses weaker PoT capabilities in smaller models and introduces a self-induced PoT variant, it does not address tasks fundamentally unsuited to programmatic reasoning (e.g., tasks that do not decompose naturally into executable programs or symbolic forms), leaving the framework’s broader applicability unclear.

**Missing comparison with advanced reasoning frameworks:**
The paper does not discuss or compare against other recent multi-step reasoning paradigms such as ReAct, Reflexion, or tool-augmented reasoning, which might also address efficiency and correctness issues at inference time.

**Limited generalization analysis:**
The Bayesian model depends on seed probabilities $(c, a_1, a_2)$ estimated from data. For the data-independent variant, the method assumes $a_2 \approx 1$ (i.e., agreement almost guarantees correctness). It is not demonstrated whether these parameters—or this assumption—generalize across diverse tasks or models.

**Questions:**

**1. Applicability:**
For problem classes that are not easily representable as executable programs, what is the expected behavior of the CoT–PoT ensembling framework? Do you observe failure modes where PoT is systematically inapplicable or misleading, and how should practitioners detect or mitigate such cases?

**2. Generalization of the Bayesian model:**
The Bayesian scheme uses seed probabilities estimated from held-out data. Can these learned parameters $(c, a_1, a_2)$ be transferred between datasets, tasks, or model families? If not, how sensitive is method performance to mismatches between training-held-out data and test distributions, and do you have practical recommendations for re-estimating or adapting these priors in low-data regimes?

**3. Consistent error reduction:**
Have you evaluated whether cross-modal agreement reduces the incidence of self-consistent but incorrect outputs (i.e., situations where multiple samples agree on a wrong answer)? In particular, can you relate your empirical findings to the phenomenon discussed in “Too Consistent to Detect: A Study of Self-Consistent Errors in LLMs”—does cross-modal ensembling mitigate or merely shift such failure modes?

---

> ### Author Response · Authors · 2025-11-18
> **Response to Review FMp9 [1/2]**
>
> We thank the reviewer for their very valuable feedback.
>
> **W1 (PoT applicability / decomposability)**
>
>  We agree that the benefits of our approach are strongest for tasks that admit some form of programmatic or symbolic decomposition (which includes most mathematical reasoning tasks). This aligns with the large class of tasks targeted by existing program-based and tool-augmented reasoning methods in the literature (e.g. Tora, ReTool, etc.). Our method is designed for and best suited to this broad category. For tasks that are not naturally expressible as programs, the PoT channel will contribute less signal, but the framework still functions: PoT samples may still provide partial structure (e.g., arithmetic steps, helper variables) and non-programmatic reasoning may still be done in commented code, and disagreement prevents early stopping. We acknowledge that such non-programmatic tasks lie outside the main scope of our experiments and we have not evaluated on this, and we will clarify this scope and the intended applicability domain in the text.
>
> **W2 (Missing comparison with ReAct / Reflexion / tool-augmented reasoning)**
>
>  These approaches (ReAct, Reflexion, tool-augmented reasoning, etc.)  require multi-turn LLM calls, tool orchestration, or explicit search control, and less naturally fit into the self-consistency paradigm as single shot prompting modalities. Our goals and scope were orthogonal: given that self-consistency is a widely adopted paradigm in llm reasoning and its high computational cost is a huge challenge (with established line of work on improving efficiency),  we propose an approach that can drastically improve its efficiency while also providing higher accuracy. Hence the baselines we compare to are all self-consistency approaches in either modalities and the best early stopping self consistency approaches. We show how cross modal agreement between two general modalities can benefit self consistency, using CoT and PoT as representing two general classes of reasoning modalities (llm-only inference vs program based inference techniques). To ensure uniformity and controlled experimentation, we even ensured the same prompt examples were used for both CoT and PoT modalities to study them on equal footing rather than allowing any specialized optimizations for either modality that may cloud the benefits of cross modal agreement. Hence it was not our goal to show that our approach is the best reasoning approach compared to any other reasoning approaches outside of self-consistency (which are many).  These  other methods therefore do not serve as direct baselines, but we agree that in principle they can also be incorporated as optimized modalities in our cross modal framework. We will add discussion acknowledging these related paradigms and explaining our scope and focus in this work with respect to them.
>
> **W3 (Generalization of Bayesian assumptions).**
>
>  We agree that seed probabilities depend on the underlying model and dataset, and is why we provide both data-driven and data-independent instantiations. The strong empirical finding that across all four large models and five benchmarks (Table 1) indicates that cross-modal agreement is consistently a powerful signal in practice for the domain of mathematical reasoning tasks we consider here. Furthermore, on the weaker 3B model—where is not as extreme—the data-driven variant (CP_DAA) outperform the data-independent ones, demonstrating that this assumption may not hold in all situations and in such cases the data-driven approaches will be more robust.

---

> ### Author Response · Authors · 2025-11-18
> **Response to Review FMp9 [2/2]**
>
> **Q1 (Applicability)**
>
> We agree that not all tasks lend themselves naturally to executable program representations.  In such cases, the expected behavior of our framework is conservative rather than misleading: the PoT channel tends to produce low-consistency or low-quality outputs, which means cross-modal agreement is rarely triggered, and the method defaults toward behavior similar to standard CoT self-consistency or adaptive consistency. This is already visible in our 3B-parameter case study, where the base model PoT is very weak—yet the framework remains stable and often still beneficial. We have not observed systematic failure modes where PoT misleads the ensemble; rather, PoT simply provides little usable signal, and the Bayesian/posterior conditions required for early stopping are not met. Practitioners can detect this by monitoring simple diagnostics such as: (i) the frequency of PoT execution errors, (ii) repeated production of non-executable code, or (iii) absence of PoT–CoT agreement over several samples. In such cases, the practitioner can disable early stopping or fall back to CoT-only SC. These applicability guidelines are similar to using any popular program-aided LLM inference systems in practice (e.g. ToRA, ReTool, PAL, PoT, …).
>
> **Q2 (Generalization of the Bayesian model)**
>
> The seed probabilities are model- and task-dependent, so we do not assume perfect transfer across domains. However, in practice the key parameter a_2 (safety given agreement) is consistently close to 1 across all four large models and five datasets we tested, which is why the data-independent variant already works well. When this assumption weakens—as in our 3B case study—the data-driven variants automatically outperform the fixed version, showing the method’s robustness to mismatch.
> Estimating these parameters requires only simple agreement counts, so a small calibration set (we used 100 samples) is sufficient. We have not experimented with very low-data regimes and can consider those in future work.
>
> **Q3 (Consistent error reduction)** :
>
> We did not explicitly re-implement the metrics from “Too Consistent to Detect”, but our analysis directly targets the same phenomenon. First, the high empirical values of a_2 (safety given cross-modal agreement) across models and datasets indicate that “self-consistent but wrong” events are much rarer when we require CoT–PoT agreement than when we rely on intra-modal majorities. Our notion of “safety” itself states “either the answer is correct or the full sampling will also get the wrong answer” and we find empirically that safety *given agreement* is almost certain (a_2 = 1). Second, our ablation comparing CPFF (first CoT–PoT agreement) with AFS-C/AFS-P (first–second within-modality agreement) shows that intra-modal agreement leads to a noticeable accuracy drop, whereas cross-modal agreement maintains or improves accuracy, suggesting that cross-modal SC substantially mitigates—rather than merely shifts—these failure modes. We will add a short discussion clarifying this connection and positioning our results as complementary evidence that diversity of error modes (CoT vs PoT) is critical for avoiding “too-consistent-but-wrong” behavior.

---

> > ### Comment · Reviewer_FMp9 · 2025-11-28
> >
> > Thanks for the response.

---

### Official Review · Reviewer_ecnh · 2025-10-30

**Soundness:** 2
**Presentation:** 2
**Contribution:** 2
**Rating:** 2
**Confidence:** 5

**Summary:**

The paper introduces a method that combines CoT and PoT reasoning through majority voting. Experimental results demonstrate that this approach enhances reasoning accuracy and outperforms traditional majority voting techniques.

**Strengths:**

1. The experimental setup is well-designed and yields strong results.

2. The paper is clearly written and easy to follow.

**Weaknesses:**

1. The idea presented in the paper is not particularly novel. Prior work from two years ago has already explored combining CoT and PoT [1]. Moreover, the combination of only CoT and PoT offers limited contribution, as there are several other reasoning methods, such as ToT [2], that could be considered. It would be more valuable to investigate the integration of a broader range of reasoning approaches.


2. The PoT method is relatively outdated, and numerous recent approaches have advanced the use of code in reasoning—particularly with the emergence of reinforcement learning techniques, such as ReTool [3], as well as several deep research efforts. Given these developments, combining two older reasoning methods like CoT and PoT offers limited relevance and impact in the current landscape.

[1] Automatic Model Selection with Large Language Models for Reasoning. EMNLP 2023

[2] Tree of Thoughts: Deliberate Problem Solving with Large Language Models. NeurIPS 2023

[3] ReTool: Reinforcement Learning for Strategic Tool Use in LLMs. Arxiv 2025

**Questions:**

NA

---

> ### Author Response · Authors · 2025-11-18
>
> We thank the reviewer for their very valuable feedback.
>
> **W1 (Prior work from two years ago has already explored combining CoT and PoT)**
>
> We do not claim novelty in being the first to combine CoT and PoT (many works have proposed various combinations of these two techniques in different ways, as reviewer rightly notes and we also state in related work). Our main novel finding and contribution is that agreement between CoT and PoT provides a uniquely strong signal for **early-stopping in self-consistency (SC)** to provide a **huge efficiency improvement of 8x** overall, and in particular, inference from **only 2 samples  in 75.9% of cases**  - no prior work  has shown such a finding to our knowledge. For example, even reference [1] which reviewer mentions (“Automatic Model Selection”) requires at least 10 LLM calls in their best method that they compare to self consistency (while our method requires only 2 calls in majority of cases). Also, their approach depends on additional specialized LLM prompting calls for model selection. Our approach in contrast provides a clean incorporation of the two modalities in the SC paradigm without such additional overhead calls. SC is a widely used paradigm in LLM reasoning where computational cost of sampling is a huge burden and challenge, and improving its efficiency is an active area of research (see related work) – and ours is the first work to provide a cross modal approach to early stopping in SC, showing drastic efficiency gains while providing higher accuracy, with majority of inferences from only 2 samples.
> We can improve our framing and description to clarify this positioning to avoid any confusion. We will also include the comparison to [1] in related work, thanks.
>
> **W2 (broader range of reasoning methods)**
>
> Regarding ToT and other reasoning styles [2], we agree that numerous other reasoning approaches have been proposed in the literature. Firstly, we note that our general Bayesian framework is not inherently limited to CoT/PoT; it only assumes multiple “reasoning channels” whose agreement can be modeled. We focused on CoT and PoT in particular as they represent two widely used and general classes of reasoning techniques (direct LLM inference vs programmatic tool use). They also provide uniform single shot prompting techniques that directly fit into the SC paradigm (unlike methods like ToT that require multiple LLM calls and backtracking, etc, which make it more complicated to measure efficiency and accuracy improvements within the SC paradigm). Our goal was  to demonstrate the benefits of incorporating these two different modalities in SC in order to improve the efficiency and accuracy of SC – in fact, we ensured our prompts used exactly the same few shot examples for uniformity across both modalities in order to analyse  their combination in a controlled experimental setting.
>
> **W3 (older reasoning methods)**
>
> Our goal and scope in this work was not to produce the most advanced reasoning system, but to study in a uniform and controlled setting the benefits of combining two very general and widely used modalities in LLM reasoning  represented by CoT (direct LLM inference) and PoT (program-based inference). PoT is used  to represent the general approach of  generic program-based reasoning (the model outputs executable code whose result determines the answer), not as a commitment to a specific outdated method. In fact, we wanted to explicitly avoid any advanced optimizations in either modality so we can uniformly examine the benefits of cross-modal agreement in a controlled experimental setting: which is why our prompts are exactly uniform and use the same few shot examples for both modalities. Hence we avoided highly specialized implementations in either modality to put them on *equal footing* and avoid clouding the benefits of cross-modal agreement.
>
> However, in general our cross-modal SC framework is agnostic to what specific modalities are used:  it could be a more recent code-centric model or an RL-enhanced tool-using agent such as ReTool [3]. In fact, in practice we would expect and advise that one should choose the most advanced state of the art systems for the different modalities to get the most optimal performance.  In all these cases, our Bayesian agreement mechanism and early-stopping strategy remain applicable. We will revise the related-work and discussion sections to make this orthogonality explicit and to emphasize that our framework can incorporate more advanced code-based or RL-trained reasoning systems as the PoT channel.

---

### Official Review · Reviewer_bUyj · 2025-10-31

**Soundness:** 2
**Presentation:** 3
**Contribution:** 2
**Rating:** 2
**Confidence:** 5

**Summary:**

This paper proposes to combine chain-of-thought and program-of-thought as self-consistency measure. Experiments indicate that ensembling CoT-PoT improves accuracy, and is a more efficient approach.

**Strengths:**

N/A The paper writing is clear, experiments are in general thorough.

**Weaknesses:**

Unfortunately, I believe this paper is currently below the acceptance bar for the conference. Rather than continuing to refine it for submission, I would suggest that the authors consider discontinuing this project or substantially rethinking its core idea.

First, the combination of CoT and PoT reasoning is a rather straightforward extension, and similar attempts have been explored several years ago. Conceptually, PoT does not provide a complementary signal to CoT. That is to say, if a problem cannot be solved by PoT, it is unlikely that CoT alone would succeed either.
Second, the experimental results show marginal improvements from combining CoT and PoT compared with existing approaches, validating what my understanding of this project.

**Questions:**

N/A

---

> ### Author Response · Authors · 2025-11-18
>
> We thank the reviewer for their very valuable feedback.
>
> **W1 (combination of CoT and PoT reasoning has been explored several years ago.)**
>
> Our main contribution  is not a combination of CoT-PoT (which prior works have done as reviewer notes), but showing that cross-modal agreement provides a uniquely **strong early-stopping signal in the self-consistency (SC) framework**, enabling drastic efficiency gains:  requiring **only 2 samples in 75.9% of cases** and **8x  efficiency gain** overall. Self-consistency is a widely used technique in LLM reasoning but its huge sampling cost is a major challenge and there is a continuing line of work to improve its efficiency (see related works). None of these prior works have achieved inference from *only two samples* or such huge efficiency gains overall while providing better accuracy, and we show how this can be robustly done with a cross-modal approach.
>
> To clarify, we do not claim key novelty in being the first to combine CoT and PoT – which  prior works have done (for different reasons and in different ways) as reviewer states and we also describe in related work (though our particular combination techniques and **Bayesian model for early stopping** have not been proposed before). We also do not claim a major contribution on increase in accuracy from CoT-PoT (which is incremental and also shown in prior methods as reviewer notes). Our main contribution is the drastic improvement in efficiency that can be achieved by early stopping with CoT-PoT.. We believe this finding not only has significant practical impact but also shows how cross-modal agreement can robustly avoid the need for statistical  sampling with LLMs for mathematical reasoning tasks – which we believe is an interesting finding for this  research community.
> We will work on clarifying and effectively positioning this central contribution to avoid confusion about the key finding and benefit our technique is actually offering.
>
> **W2 (if a problem cannot be solved by PoT, it is unlikely that CoT alone would succeed either)**
>
> We are not sure what exactly the reviewer means by this - our extensive evaluation shows consistently that there are many cases where PoT fails and CoT succeeds (see cases where SC_cot has higher accuracy than SC_pot in Table 2). The point is not whether a problem **can** potentially be solved by PoT or CoT, but that LLMs tend to make *different kinds of errors* when solving with PoT or CoT (the error modes of the two modalities are different). E.g. PoT errors are in symbolic formulation (like using wrong operator syntax in code) while CoT may make calculation errors. Hence when there is agreement between modalities we achieve high confidence to stop sampling early. We can clarify this point further in the introduction.
>
> **W3 (experimental results show marginal improvements from combining CoT and PoT)**
>
> There may be confusion on this that we would like to clarify: the marginal results are only for **accuracy** improvement (1.6% over standard  SC) which is not our main contribution but a minor secondary contribution and a prerequisite for our main results on early stopping. For our main contribution of **efficiency** with early stopping the results  are by no means marginal: we show a huge 8x reduction in the number of samples over SC and that inference can be done with only 2 samples in 75.9% of cases (no prior early stopping method can show inference from only 2 samples in any case).

---

> > ### Comment · Reviewer_bUyj · 2025-11-26
> >
> > Thanks for your comments!

---

### Note · Authors · 2026-01-05

I have read and agree with the venue's withdrawal policy on behalf of myself and my co-authors.